# Therapeutic role of miR-19a/19b in cardiac regeneration and protection from myocardial infarction

Feng Gao[1], Masaharu Kataoka[2,3], Ning Liu[1], Tian Liang[1], Zhan-Peng Huang[2,4], Fei Gu[2], Jian Ding[2], Jianming Liu[2], Feng Zhang[1], Qing Ma[2], Yingchao Wang[5], Mingming Zhang[2], Xiaoyun Hu[2], Jan Kyselovic[6], Xinyang Hu[5], William T. Pu [2,7], Jian'an Wang[5], Jinghai Chen [1] & Da-Zhi Wang[2,7]

The primary cause of heart failure is the loss of cardiomyocytes in the diseased adult heart. Previously, we reported that the *miR-17-92* cluster plays a key role in cardiomyocyte proliferation. Here, we report that expression of miR-19a/19b, members of the *miR-17-92* cluster, is induced in heart failure patients. We show that intra-cardiac injection of miR-19a/19b mimics enhances cardiomyocyte proliferation and stimulates cardiac regeneration in response to myocardial infarction (MI) injury. miR-19a/19b protected the adult heart in two distinctive phases: an early phase immediately after MI and long-term protection. Genome-wide transcriptome analysis demonstrates that genes related to the immune response are repressed by miR-19a/19b. Using an adeno-associated virus approach, we validate that miR-19a/19b reduces MI-induced cardiac damage and protects cardiac function. Finally, we confirm the therapeutic potential of miR-19a/19b in protecting cardiac function by systemically delivering miR-19a/19b into mice post-MI. Our study establishes miR-19a/19b as potential therapeutic targets to treat heart failure.

[1] Department of Cardiology, The Second Affiliated Hospital, Institute of Translational Medicine, Zhejiang University School of Medicine, 310029 Hangzhou, China. [2] Department of Cardiology, Boston Children's Hospital, Harvard Medical School, 300 Longwood Avenue, Boston, MA 02115, USA. [3] Department of Cardiology, Keio University School of Medicine, Tokyo 160–8582, Japan. [4] Center for Translational Medicine, The First Affiliated Hospital, NHC Key Laboratory of Assisted Circulation, Sun Yat-sen University, Guangzhou 510080, China. [5] Department of Cardiology, The Second Affiliated Hospital, Zhejiang University School of Medicine, 310009 Hangzhou, China. [6] Faculty of Pharmacy, Comenius University, Bratislava 832 32, Slovak Republic. [7] Harvard Stem Cell Institute, Harvard University, Cambridge, MA 02138, USA. Correspondence and requests for materials should be addressed to J.C. (email: jinghaichen@zju.edu.cn) or to D.-Z.W. (email: Da-Zhi.Wang@childrens.harvard.edu)

During heart development, the specification of cardiac progenitor cells and cardiomyocytes, their proliferation and differentiation, and ultimately the maturation of terminally differentiated cardiomyocytes is a complicated process dictated by the genetic blueprint and orchestrated by regulatory networks involving transcriptional and epigenetic control of gene expression. It is generally believed that terminally differentiated cardiomyocytes in adult mammalian hearts have exited the cell cycle and have lost the ability to proliferate[1]. As a result, adult hearts fail to regenerate themselves upon myocardial infarction (MI) (also referred to as heart attack) in which massive numbers of cardiomyocytes are lost[2]. Despite numerous efforts in the past decades to promote cardiovascular health, heart failure following MI remains the leading cause of mortality and morbidity in humans. Interestingly, recent work has begun to reveal that postnatal mammalian hearts possess the potential for regeneration under a variety of pathophysiological conditions[3–6], raising hope for a therapeutic approach to repair damaged hearts.

microRNAs (miRNAs), a class of ~21–23 nucleotide non-coding RNAs, have emerged as key regulators of cardiomyocyte proliferation, differentiation, and cardiac function. Previously, functional screening identified many miRNAs that have the potential to stimulate neonatal cardiomyocyte proliferation[7]. Among them, has-miR-199a-3p and hsa-miR-590-3p, upon intra-cardiac injection, were shown to stimulate cardiomyocyte proliferation and improve cardiac function in response to MI[7]. Intriguingly, Aguirre et al. reported that miR-99/100 and let-7 modulate cardiac regeneration in zebrafish and mouse; inhibition of these miRNAs stimulated cardiac regeneration, apparently by enhancing cardiomyocyte dedifferentiation[8,9]. In another study, the function of miRNAs in regulating cardiomyocyte proliferation and heart regeneration was linked to the Hippo/Yap pathway, in which members of the *miR302-367* cluster directly target key components of the Hippo/Yap pathway[10]. miR-34a, which was initially demonstrated to regulate the ageing process, is an important regulator of cardiomyocyte proliferation and cardiac regeneration such that inhibition of this miRNA leads to enhanced cardiomyocyte proliferation and cardiac regeneration in response to MI[11]. Intriguingly, a recent study showed that a single-dose intracardiac injection of miR-199a-3p and miR-590-3p mimics was able to protect cardiac function in response to MI, underscoring the therapeutic potential of miRNAs in cardiomyopathy[12].

The *miR-17-92* cluster of miRNAs has been shown to regulate animal development and cell proliferation[13–16]. It has also been shown that individual members of the *miR-17-92* cluster play distinct yet cooperative roles[16,17]. Using mouse models of genetic mutation and overexpression, we have previously reported that the *miR-17-92* cluster plays a critical role in cardiomyocyte proliferation in embryonic, postnatal, and adult hearts. Cardiomyocyte-specific overexpression of miR-17-92 enhanced cardiomyocyte proliferation in transgenic mice and protected the heart from MI. In particular, we showed that miR-19a and miR-19b are necessary and sufficient for cardiomyocyte proliferation in vitro in isolated neonatal rat cardiomyocytes[18]. Here, we explore the therapeutic potential of miR-19a/19b in cardiac regeneration and protection. We report that intra-cardiac injection or systemic delivery of miR-19a/19b mimics or using an adeno-associated virus 9 (AAV9) delivery method to overexpress miR-19a/19b in mouse heart reduces MI-induced cardiac injury and preserved cardiac function. Our results suggest that miR-19a/19b could become therapeutic targets to prevent and treat cardiac disease.

## Results

**miR-19a/19b is dysregulated in diseased hearts**. We examined the expression of miR-19a/19b and found that these miRNAs are highly expressed in the hearts of neonatal mice (Fig. 1a). Their expression levels are decreased in postnatal day 12 mouse hearts (Fig. 1a), consistent with the view that these miRNAs are involved in cell proliferation. We further investigated the distribution of these miRNAs in cardiomyocytes and non-cardiomyocytes and found that miR-19a is enriched in cardiomyocytes of adult mouse hearts (Fig. 1b).

Next, we studied the expression of miR-19a and miR-19b in mouse models of cardiomyopathy. The expression of both miR-19a and miR-19b was increased in the heart 3 days after MI (Fig. 1c). Two weeks after MI, the expression levels of these miRNAs had returned to normal; however, their expression level increased again 4 weeks after MI (Fig. 1d). We then investigated whether the expression of miR-19a/19b is altered in another mouse model of cardiac remodeling, transverse aortic constriction (TAC)-induced cardiac hypertrophy. However, we found that the expression of miR-19a and miR-19b was not changed in hypertrophic hearts 3 days or 2 weeks after TAC. As a control, we showed that the expression of miR-21 increased in hypertrophic hearts (Fig. 1e), consistent with previous reports[19,20]. These results indicate that the expression (and function) of these miRNAs is specifically associated with MI, not cardiac hypertrophy.

To investigate clinical relevance, we asked whether the expression of these miRNAs is altered in the hearts of human patients with cardiovascular diseases. For this purpose, we performed qRT-PCR assays and found that the expression levels of primary transcripts of the miR-17-92 cluster, which includes miR-19a, were increased in patients with dilated cardiomyopathy (DCM) and coronary artery disease (CAD) (Fig. 1f). As a control, we confirmed that the expression of cardiomyopathy marker gene atrial natriuretic factor (ANF) was increased in DCM but not CAD heart samples (Fig. 1f). Together, these results indicate that the expression of miR-19a and miR-19b is dynamically regulated in cardiomyopathy.

**miR-19a/19b protects the heart from MI**. To test the function of miR-19a/19b in the heart, we directly injected miRNA mimics into the heart of a mouse model of MI. The left anterior descending (LAD) coronary artery was permanently ligated (LAD-ligation) in adult mice to create a MI and heart failure model. miR-19a and miR-19b-1, which share an identical seed sequence, are two members of the miR-19 family encoded by the miR-17-92 cluster (Fig. 2a). We first tested whether miR-19a and miR-19b possess similar functions in the heart. miR-19a or miR-19b mimics were directly injected into the heart (intra-cardiac) at regions adjacent to the ligation site. Indeed, injection of either miR-19a or miR-19b mimics preserved cardiac function at 5 days and at 2, 7, and 9 weeks post-MI (Fig. 2b; Supplementary Table 1). Injected hearts exhibited reduction of scar size at 2 months after MI (Fig. 2c, d; Supplementary Table 1). Given that miR-19a and miR-19b have almost identical RNA sequences and both produce similar cardiac protection in infarcted hearts, we decided to combine miR-19a and miR-19b in the following experiments to systematically determine the role of miR-19 in cardiac function and regeneration.

We next carefully examined the pharmacokinetics of miRNA mimics after intracardiac injection and MI. We found that most miRNA mimics appear to be located outside of cell membrane of cardiomyocytes 2 h after injection; 6 h after injection, they are located in the cytoplasm, and at 12–24 h they are located in the whole cell. 48 h after injection, we still detect homogenous distribution of miRNA mimics in cardiomyocytes (Supplementary Fig. 1). Increased miR-19a/19b expression was still detected by qRT-PCR in the heart 4 days after injection. However, no

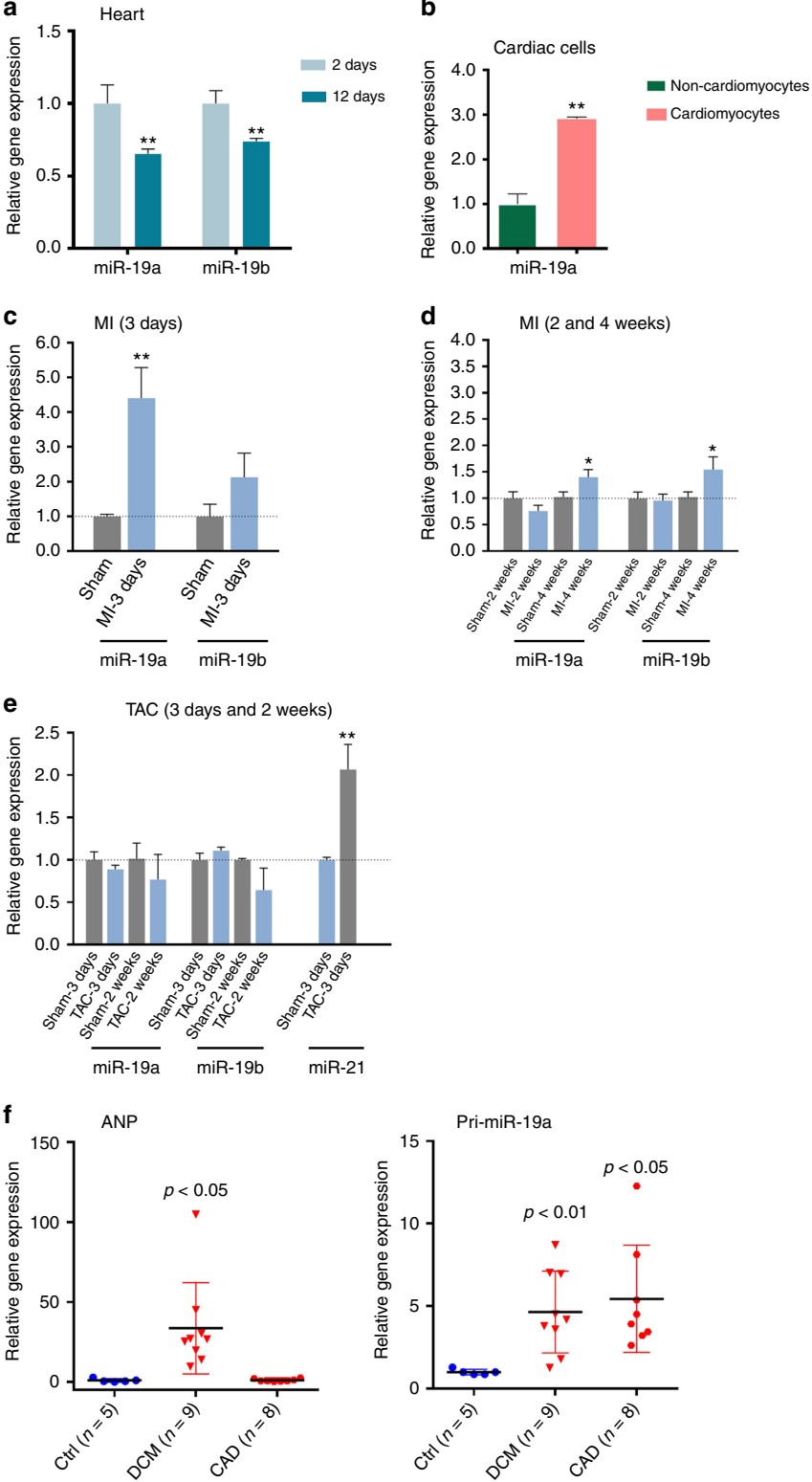

**Fig. 1** Expression of mir-19a/miR-19b in the heart and cardiomyocytes. **a** qRT-PCR of miR-19a and miR-19b at 2 and 12 days postnatal mouse hearts. $n = 3$ mice. **b** In isolated adult cardiomyocytes and non-cardiomyocytes, qRT-PCR detection of the expression of miR-19a. $n = 3$ mice. **c**, **d** In mouse myocardial infarction (MI) model, qRT-PCR detection of the expression of miR-19a and miR-19b (**c**) at 3 days post-MI ($n = 6$ mice) and (**d**) at 2 weeks and 4 weeks post-MI. $n = 3$–6 mice. **e** In mouse transverse aortic constriction (TAC) induced cardiac hypertrophy model, qRT-PCR detection of the expression of miR-19a, miR-19b, and miR-21 at 3 days and 2 weeks post-TAC. Expression of miR-21 was considered as positive control. $n = 2$–3 mice. **f** In human heart disease samples, qRT-PCR detection of the expression of pri-mir19a in hearts with dilated cardiomyopathy (DCM) and coronary artery disease (CAD). The expression of ANF (encoded by *Nppa*) was considered as positive control. All panels, statistical significance was calculated using Student's *t*-test and data are presented as means ± s.e.m. *$p < 0.05$, **$p < 0.01$ vs. control. Source data are provided as a Source Data file

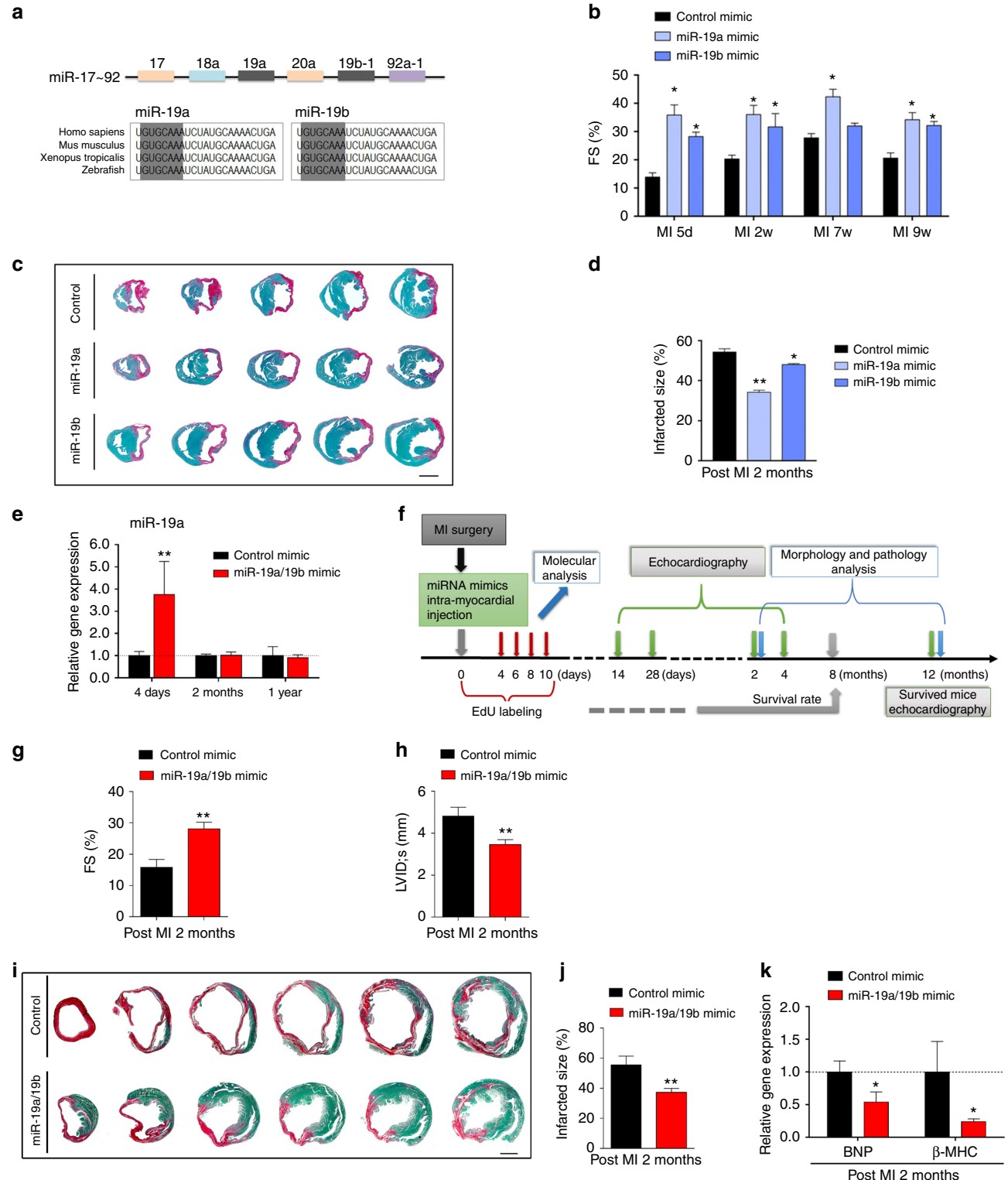

increased miR-19a/19b expression was detected 2 months and 1 year later in miR-19a/19b mimic-injected hearts (Fig. 2e). We systematically analyzed cardiac phenotypes in these miR-19a/19b mimic-injected mice (Fig. 2f; Supplementary Table 2). Cardiac function was measured using echocardiography, and we found that mice injected with miR-19a/19b mimics exhibit improved cardiac function, as evidenced by preserved fractional shorting (FS%), when compared with control mimic-injected mice (Fig. 2g; Supplementary Table 2). Interestingly,

overexpression of miR-19a/19b also prevented MI-induced cardiac dilation in these mice (Fig. 2h; Supplementary Table 2). We performed histological analysis and found that injection of miR-19a/19b mimics significantly reduced scar formation in infarcted hearts (Fig. 2i). Quantification of infarct size confirmed this observation (Fig. 2j). We further examined the expression of molecular marker genes and found that the expression of brain natriuretic peptide (*Nppa*) and beta-myosin heavy chain (β-MHC, encodes by *Myh7*), molecular markers of

**Fig. 2** Direct injection of miR-19a/19b mimics protects the heart from myocardial infarction. **a** Gene structure of the *miR-17–92* cluster and conserved sequences of miR-19a and miR-19b across species. Seed sequences are highlighted. **b** Echocardiography analyses of mice with intra-cardiac injection of individual miR-19a, miR-19b or control mimics at 5 days, 2, 7, and 9 weeks post MI. $n = 3–6$ mice. **c** Transverse sections of miR-19a, miR-19b or control mimic injected hearts at 2 months post MI. Sirius red/fast green marks myocardium (green) and scar (red). Scale bar = 2 mm. **d** Quantification of the size of scar. $n = 3–4$ mice. **e** qRT-PCR of miR-19a in heart at 4 days, 2 months, and 1 year after intra-cardiac injection of miR-19a/19b mimics. $n = 3–5$ mice. **f** Experimental design. Mice receiving miR-19a/19b mimic post-MI were assessed for cardiac function short term and long term, as well as morphological assessment. **g** Left ventricular fractional shortening (FS%) and (**h**) LV internal dimension at end-systole (LVID;s). $n = 5$ mice in control mimic group, $n = 10$ mice in miR-19a/19b mimic group. **i** Representative images of series of transverse sections after injection of miR-19a/19b mimics compared to control group at 2 months after MI. Sirius red/fast green collagen staining marks myocardium (green) and scar (red). Scale bar = 2 mm. **j** Quantification of the size of scar in the hearts after injection of miR-19a/19b mimics. $n = 5$ control mice; $n = 4$ miR-19a/19b mimic-treated mice. **k** qRT-PCR detection of expression of pathological remodeling marker genes BNP (encoded by *Nppb*) and β-MHC (encoded by *Myh7*). $n = 5$ control mice; $n = 6$ miR-19a/19b mimic-treated mice. Statistical significance was calculated using Student's *t*-test in **b**, **d**, **e**, **g**, **h**, **j**, and **k** and data are presented as means ± s.e.m. *$p < 0.05$, **$p < 0.01$ vs. control. Source data are provided as a Source Data file

cardiomyopathy, was reduced 2 months after miR-19a/19b injection (Fig. 2k). These results indicate that miR-19a/19b protects the heart from MI.

**Early and late cardiac protection by miR-19a/19b**. Interestingly, we observed that miR-19a/19b-mediated cardiac protection exhibited two apparently distinct phases: an early phase immediately after MI (early) and a later phase (late, or long-term) (Fig. 3a). More than 50% of mice injected with control miRNA mimics died within a week after MI, due to heart failure; in contrast, only about 20% died when miR-19a/19b mimics were injected (Fig. 3a). A similarly high mortality rate upon MI was reported previously in a similar study[12,21]. When we applied less severe infarction by lowering the ligation position on the LAD during MI surgery, we observed reduced overall lethality, and injection of miR-19a/19b mimics still exhibited beneficial effects (Supplementary Fig. 2A). We examined cardiac function in mice injected with miR-19a/19b or control mimics at different time points and found that injection of miR-19a/19b mimics protects cardiac function at 2–4 weeks and 4 months, when compared with controls (Fig. 3b; Supplementary Table 2). Intriguingly, we observed that one-time injection of miR-19a/19b mimics during the procedure of LAD ligation was able to preserve cardiac function up to 12 months after MI (Fig. 3c; Supplementary Table 2), consistent with a similar recent report showing that miR-199a-3p and miR-590–3p mimics were able to protect the heart from MI injury[12]. Histological analysis revealed that miR-19a/19b mimic injected hearts displayed reduced infarcted size (Fig. 3d, e), supporting the view that these miRNAs improved cardiac function. Given that overexpression of miR-19a/19b substantially reduced the scar size after MI (Figs. 2i, 3d), we asked whether the expression of genes related to fibrosis was repressed by miR-19a/19b. As expected, the expression levels of collagen genes *Col1a1*, *Col3a1*, elastin (*ELN*), and fibrillin 1 (*FBN1*) and transforming growth factor beta receptor 2 (*TGFBR2*) were all reduced in miR-19a/19b mimic-injected hearts (Fig. 3f). Twelve months after miR-19a/19b injection, we observed decreased expression of Col3a1, as well as BNP (encoded by *Nppb*) and β-MHC (encoded by *Myh7*) (Fig. 3g).

**miR-19a/19b stimulates adult cardiomyocyte proliferation.** Previously, we showed that transgenic overexpression of the *miR-17-92* cluster enhanced cardiomyocyte proliferation, whereas genetic deletion of this cluster resulted in a decrease in cardiomyocyte proliferation in response to MI[18]. We further showed that introduction of miR-19a/19b mimics into isolated neonatal rat ventricle cardiomyocytes (NRVM) was sufficient to induce their proliferation in vitro[18]. In order to determine whether miR-19a/19b mimics in adult mouse hearts are able to activate cardiomyocyte proliferation, we examined EdU incorporation in cardiomyocytes in mouse hearts after miR-19a/19b or control mimic injection and MI. We detected increased EdU+ signal in cardiomyocytes after miR-19a/19b mimic injection when compared to control (Fig. 4a). Quantification revealed a slight increase in cardiomyocyte proliferation 4 days after miR-19a/19b injection (Fig. 4b); this increase became substantial 2 months later (Fig. 4c). To verify this observation, we examined the expression of phospho-Histone H3 (pH3) and found a similarly increased pH3 signal in mouse heart after miR-19a/19b mimic injection and MI (Fig. 4d). Quantification confirmed the role of miR-19a/19b in stimulating cardiomyocyte proliferation (Fig. 4e). We examined the cytokinesis marker Aurora B in control and miR-19a/19b mimic injected hearts and found that miR-19 induced the expression of Aurora B in cardiomyocytes (Figure 4f, g), suggesting the induction of adult cardiomyocyte proliferation by miR-19. In addition, the expression levels of the cell cycle-related genes cyclin B1 (*CCNB1*), cyclin D1 (*CCND1*), and cyclin-dependent kinase 1 (*CDK1*) were all elevated in miR-19a/19b mimic injected hearts after MI (Fig. 4h).

Next, we dissociated adult cardiomyocytes from hearts 3 weeks after miR-19a/19b or control mimic injection and MI (Fig. 4I). Quantification analysis confirmed an increase in total number of cardiomyocytes in miR-19a/19b mimic injected hearts (Fig. 4j). We stained isolated adult cardiomyocytes with CONNEXIN 43, α-ACTININ, and DAPI to determine the integrity and nucleation status of cardiomyocytes (Fig. 4k) and we found that injection of miR-19a/19b mimics results in an increase of mononuclear cardiomyocytes and a decrease of binucleated cardiomyocytes (Fig. 4l). Together, these results demonstrate that miR-19a/19b stimulates cardiomyocyte proliferation in adult hearts in response to MI.

**miR-19a/19b regulates cell death and inflammation pathways.** The early cardiac protection granted by miR-19a/19b immediately after MI prompted us to test whether miR-19a/19b might inhibit MI-induced apoptosis and the inflammatory response that is detrimental to cardiac regeneration. Using TUNEL assay, we assessed apoptosis in infarcted hearts injected with either miR-19a/19b or control mimics, and we found a substantial decrease in TUNEL signals in miR-19a/19b injected hearts (Fig. 5a). Quantification verified that apoptosis was decreased in both cardiomyocytes (Fig. 5b) and non-cardiomyocytes (Fig. 5c). This observation is further supported by the decreased expression of *Bim* and *Pten*, two genes indicated in the apoptosis pathway (Fig. 5d). Western blot analysis verified that BIM and PTEN protein levels are decreased in miR-19a/19b injected hearts (Fig. 5e, f). Interestingly, both *Bim1* and *Pten* genes have previously been identified as direct targets of miR-19a/19b[22,23], suggesting that miR-19a/19b inhibits apoptosis by directly repressing apoptosis-related genes. This conclusion is further

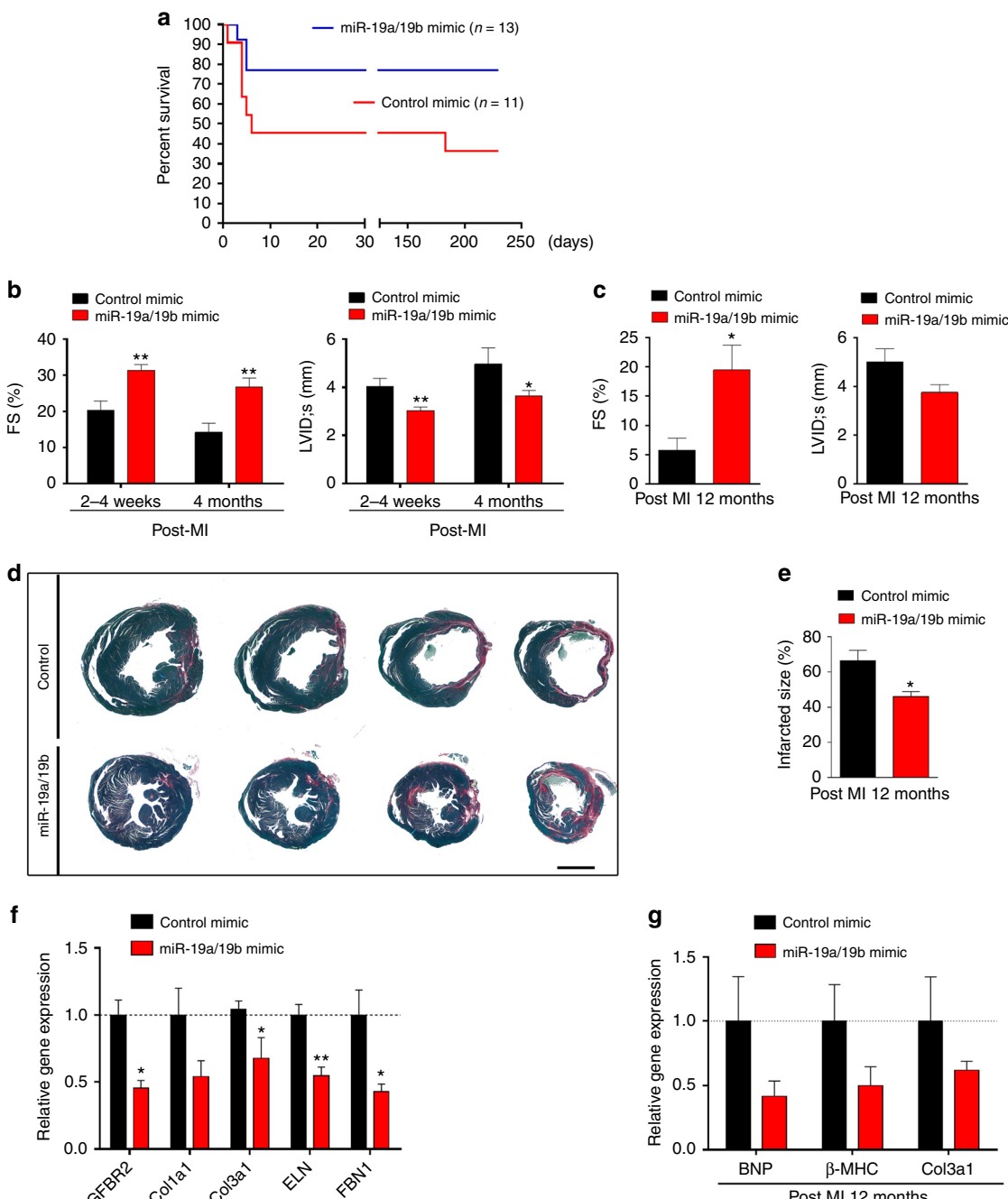

**Fig. 3** Short-term and long-term cardiac protection after myocardial infarction and injection of miR-19a/19b mimics. **a** Kaplan–Meier survival curves after injection of miR-19a/19b mimics compared to injection of control mimic after MI injury. $n = 11$ mice mimic control; $n = 13$ mice, miR-19a/19b mimics. **b**, **c** Echocardiography analyses of cardiac function after miR-19a/19b mimic injection at both short term of 2–4 weeks and 4 months (**b**) and long term of 12 months (**c**) after MI injury compared to their control group. FS% left ventricular fractional shortening. LVIDs LV internal dimension at end-systole. $n = 3$–12 mice. **d** Representative images of series of transverse sections after injection of miR-19a/19b mimics compared to control mimic at long term of 12 months after MI injury. Sirius red/fast green collagen staining marks myocardium (green) and scar (red). Scale bar = 2 mm. **e** Quantification of the size of scar after injection of miR-19a/19b mimics compared to control mice at long term of 12 months after MI injury. $n = 3$ mice. **f** qRT-PCR detection of expression of fibrotic remodeling marker genes 4 days after mimic injection and MI injury. $n = 3$–7 mice. **g** qRT-PCR detection of expression of pathological remodeling marker genes 12 months after mimic injection and MI injury. $n = 4$–6 mice. Statistical significance was calculated using Student's $t$-test in **b**, **c**, **e**, **f**, and **g** and data are presented as means ± s.e.m. $*p < 0.05$, $**p < 0.01$ vs. control. Source data are provided as a Source Data file

supported by the observation that cleaved caspase 3 levels are substantially reduced in miR-19a/19b mimic injected hearts (Fig. 5e, f). However, we did not detect significant differences between miR-19 and control mimic injected hearts when triphenyltetrazolium chloride (TTC) staining was applied to quantify nonviable myocardium, nor did we notice any difference in cardiac function in these mice 24 h after MI and miRNA mimic injection (Supplementary Fig. 2).

To better understand the molecular mechanisms underlying miR-19a/19b-mediated cardiac protection, we performed next generation RNA-sequencing (RNA-seq) in miR-19a/19b mimic and control mimic injected heart samples to obtain an unbiased,

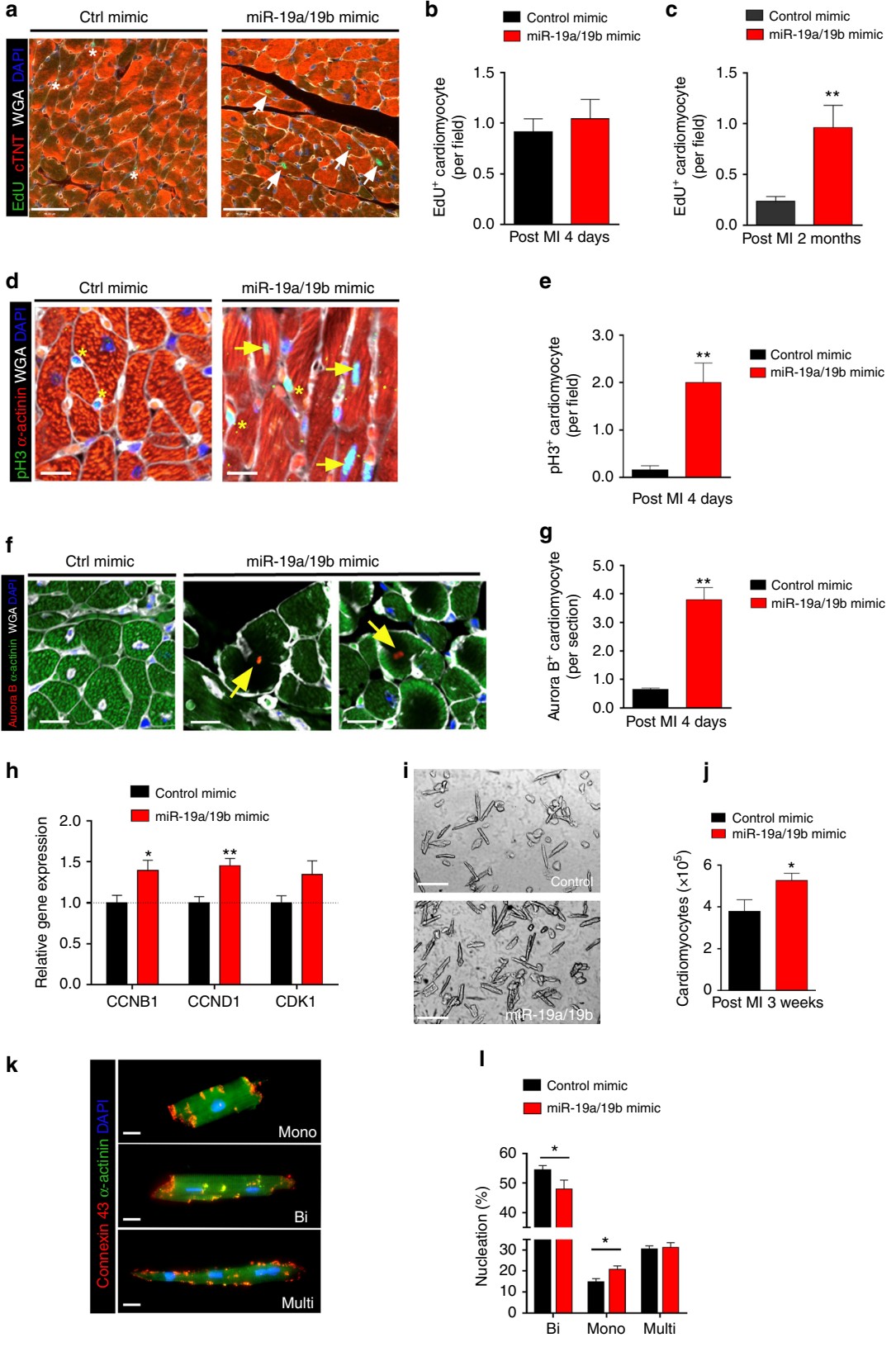

genome-wide view. We examined alterations in gene expression 4 days after MI and miR-19a/19b mimic injection. Compared to control samples, a total of 544 genes were dysregulated in miR-19a/19b mimic injected hearts with statistical significance. Among them, 393 were down-regulated and 151 up-regulated

(Fig. 5g). Gene Ontology (GO) analysis revealed that genes related to immune response, defense response, and T cell activation are enriched among the 393 down-regulated genes (Fig. 5h). Additionally, genes related to extracellular matrix, collagen network, and integrin signaling pathway are enriched

**Fig. 4** miR-19a/19b promotes cardiomyocyte proliferation after myocardial infarction. **a** Immunofluorescence staining of EdU incorporation on transverse sections of adult hearts injected with miR-19a/19b or control mimic 2 months post MI injury. EdU labels proliferating cells (green); cardiac troponin T (cTNT) marks cardiomyocytes (red); wheat germ agglutinin (WGA) marks cell surfaces (white) and DAPI labels nuclei (blue). Arrows point to EdU-positive signal in cardiomyocytes and stars mark EdU-positive signal in non-cardiomyocytes. Scale bar = 48 μm. **b, c** Quantification of percentages of EdU-positive cardiomyocytes. $n = 5$ hearts for control group, $n = 3$ hearts for miR-19a/19b mimics group, 25–30 fields per heart for each group. **d** Immunofluorescence staining of pH3 on transverse sections of adult hearts injected with miR-19a/19b or control mimic 4 days post MI injury. pH3 (green); α-actinin (red); WGA (white) and DAPI (blue). pH3-positive signal in cardiomyocytes (arrows) and non-cardiomyocytes (asterisk) are marked. Scale bar = 20 μm. **e** Quantification of pH3-positive cardiomyocytes. $n = 3$ hearts for each group, 6–8 fields per heart for each group. **f** Immunofluorescence staining of Aurora B adult hearts. Aurora B (red); α-ACTININ (green); WGA (white); and DAPI (blue). Arrows point to Aurora B-positive signals in cardiomyocytes. Scale bar = 20 μm. **g** Quantification of Aurora B-positive cardiomyocytes. $n = 4$ hearts for each group, five sections for each heart. **h** qRT-PCR of cell cycle marker genes. $n = 4$–5 hearts. **i** Representative images of cardiomyocytes isolated from adult hearts injected with control or miR-19a/19b mimic 3 weeks post MI injury. Scale bars = 100 μm. **j** Quantification of the number of cardiomyocytes. Approximately $4 \times 10^3$ cardiomyocytes were counted per group, using eight independent heart samples. **k** Immunostaining of isolated cardiomyocytes with Connexin 43 (red), α-actinin (green), and DAPI (blue). Scale bars = 10 μm. **l** For nucleation, ~$1 \times 10^3$ cardiomyocytes were counted per sample, using nine control hearts and six miR-19a/19b mimic-treated hearts. Statistical significance was calculated using Student's $t$-test in **b**, **c**, **e**, **g**, **h**, **j**, and **l** and data are presented as means ± s. e.m. *$p < 0.05$, **$p < 0.01$ vs. control. Source data are provided as a Source Data file

among miR-19a/19b down-regulated genes (Fig. 5h). In contrast, there is no GO term enrichment for the 151 up-regulated genes. We performed quantitative RT-PCR (qRT-PCR) analysis and verified down-regulated genes, including genes related to immune response and fibrosis (Fig. 5i). Interestingly, we found that several genes related to cell death, such as *BCL6*, *PIK3AP1*, and *XAF1* were also down-regulated in miR-19a/19b mimic-injected hearts (Fig. 5i). We were also able to verify several up-regulated genes with statistical significance (Fig. 5i).

The down-regulation of immune response genes in miR-19a/miR-19b injected hearts prompted us to examine how miR-19 may regulate the immune response upon MI injury. Using the widely used M1 inflammatory cell markers CD80 and iNOs, we performed immunostaining of myocardium 48 and 72 h after MI and found that miR-19a/19b mimic-injected hearts display a substantial reduction in CD80 and iNOs signals (Fig. 5j, k), suggesting a reduction in M1-type inflammatory macrophages[24–26]. Conversely, the expression level of ARG-1, which marks M2-type inflammatory macrophages, is higher in miR-19/19b mimic-injected hearts (Fig. 5l). We confirmed that the expression of genes related to M1 and M2 phages are regulated by miR-19a/19b (Fig. 5m). Prior studies indicate that suppressor of cytokine signaling 1 (*SOCS1*), a predicted miR-19a/19b target, is critical for M1-type and M2-type macrophage activation and homeostasis[26]. Indeed, we found that expression levels of both SOCS1 mRNA and protein are reduced in miR-19a/19b mimic-injected hearts (Fig. 5n, o). Accordingly, the expression level of phosphorated STAT3, a SOCS1/3 downstream mediator, was increased in miR-19a/19b injected hearts (Fig. 5n, o). Given that the Hippo/Yap pathway has been indicated in the regulation of cardiomyocyte proliferation, we examined the expression of members of the Hippo/Yap pathway genes in miR-19a/19b injected hearts. However, the expression of most of the Hippo/Yap pathway genes was not altered (Supplementary Fig. 3). Together, these results clearly demonstrate that reduction of immune response and cell death plays a key role in miR-19a/19b-mediated cardiac protection in response to MI.

**AAV-delivered miR-19a/19b protects the heart from MI**. The above results from direct injection of miR-19a/19b mimics into infarcted hearts demonstrate that these miRNAs protect the heart from MI injury. We sought to apply an independent approach to further define the function of these miRNAs in the heart. We used an AAV9 in which the cardiac-specific cTNT promoter has been incorporated to achieve cardiac-specific overexpression of miR-19a and miR-19b (AAV9-miR-19a/19b)[27,28]. An AAV9-luciferase construct was used as a control (Fig. 6a). We detected

increased miR-19a/19b expression ~72 h after AAV injection, and the increase became substantial 7 days post injection. Increased miR-19/19b expression was still detected 3 months after injection (Fig. 6b). Accordingly, we noticed no difference in cardiac function between AAV9-19a/19b and control injected hearts at 24 h. However, cardiac function is improved 7 days after AAV9-miR-19a/19b injection (Fig. 6c, d, Supplementary Table 3). We observed reductions in the expression of Caspase 3 and Bim after AAV9-miR-19a/19b injection when compared with the control group (Supplementary Fig. 4). Overexpression of miR-19a/19b using AAV9-miR-19a/19b exhibits long-term protection of the heart from MI-induced injury, as evidenced by preserved cardiac function at 2 and 3 months post MI (Fig. 6c, d; Supplementary Table 3) and reduced infarct size (Fig. 6e, f). These results support what we observed in the miR-19a/19b mimic experiments. Molecular marker analysis confirmed that AAV9-miR-19a/19b reduced the expression of cardiomyopathy markers ANF (*Nppa*), BNP (*Nppb*), and β-MHC (*Mhy7*) (Fig. 6g). We found that AAV9-miR-19a/19b also reduced the expression of fibrosis marker genes *Col1a1*, *Col3a1*, and *TGFBR2* (Fig. 6h). Consistent with previous report[18], we found that the expression levels of PTEN mRNA and protein, which is a direct miR-19a/19b target, were reduced in AAV9-miR-19a/19b-injected hearts (Fig. 6i, j, k). Together, these results confirm the role of miR-19a/19b in cardiac protection against MI.

**Therapeutic potential of miR-19a/19b in infarcted hearts**. Having demonstrated their critical role in cardiac protection in response to cardiac injury, we next asked whether miR-19a and miR-19b could serve as potential therapeutic targets to treat myocardial infarcted hearts. Six hours after MI, we delivered miR-19a/19b or control mimic formulated with MaxSuppressor in vivo RNALancerII, a lipid-based delivery reagent via tail vein injection (Fig. 7a). We carefully examined the expression of miR-19a/19b in the heart at different time points after injection, and we found that miR-19a/19b expression was readily detected in the heart 6 h after injection (Fig. 7b). Interestingly, we did not detect increased miR-19a/19b expression in control hearts without MI 24 h after injection (Fig. 7c), indicating that MI injury enhanced the uptake of miR-19a/19b in the heart. We examined cardiac function after tail vein miR-19a/19b mimic injection and found that cardiac function in miR-19a/19b mimic injected hearts was significantly improved compared to controls at 2, 6, 11, and 14 weeks (Fig. 7d, e; Supplementary Table 4). Histological analysis revealed that miR-19a/19b mimics significantly reduced scar formation in infarcted hearts (Fig. 7f). Quantification of infarct size confirmed this observation (Fig. 7g). We examined the

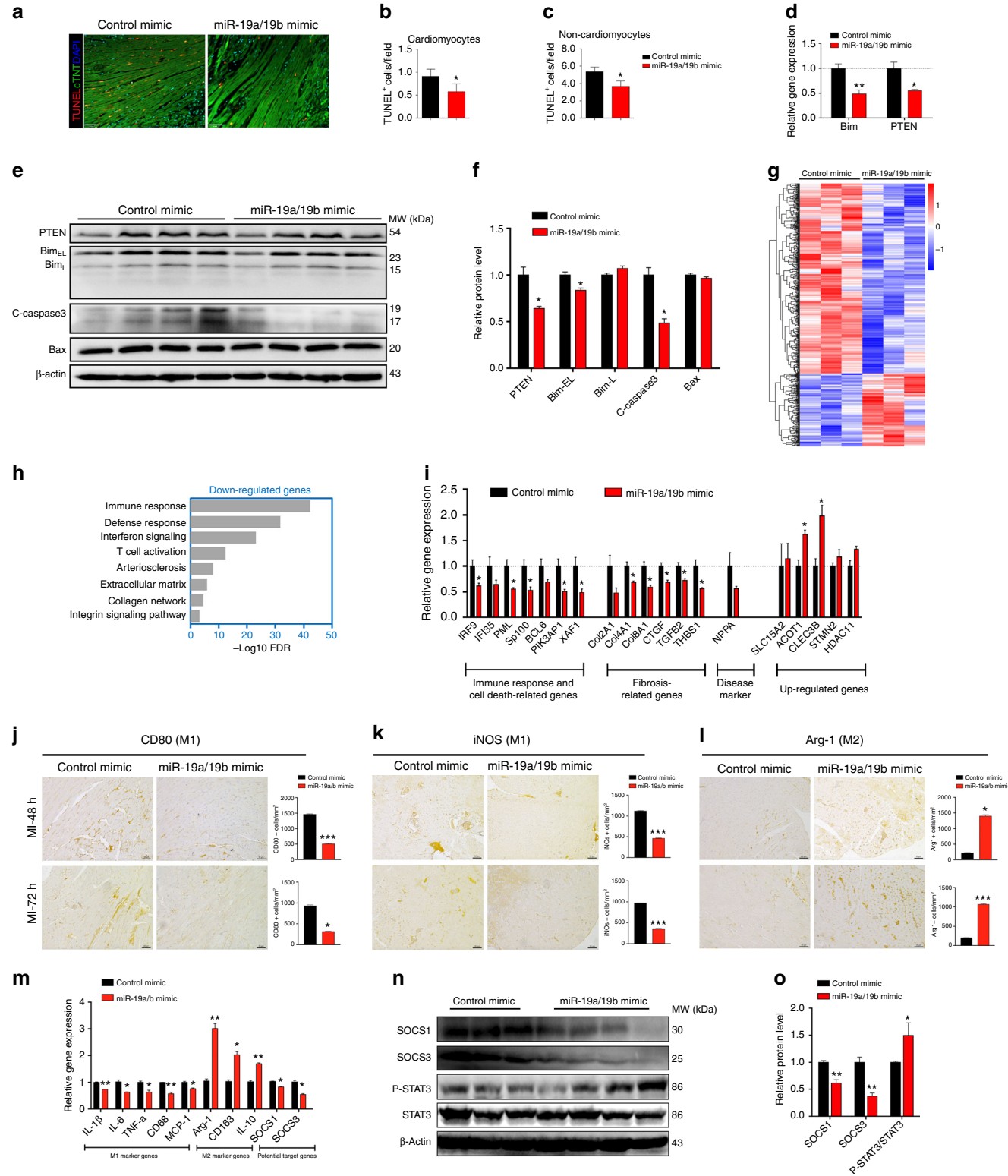

expression of SOCS1 and phosphorated STAT3 and found decreased expression of SOCS1 and increased levels of phosphorated STAT3 24 h after tail vein injection of miR-19a/19b (Fig. 7h). To further validate the therapeutic role of miR-19a/19b in cardiac protection, we utilized Lipofectamine RNAiMAX, which was recently shown to mediate miRNA delivery efficiently[12], to deliver miR-19a/19b mimics via tail vein injection.

Consistent with the findings from RNALancerII-mediated delivery, RNAiMAX-mediated systemic miR-19a/19b delivery protects the heart from MI injury, as evidenced by improved cardiac function and reduction of infarcted scar size (Supplementary Fig. 5; Supplementary Table 5). Taking together, these data indicate that miR-19a/19b are effective therapeutic targets for MI-induced heart failure.

**Fig. 5** miR-19a/19b reduces myocardial infarction-induced inflammation and cell death. **a** TUNEL staining (red) on transverse sections of adult hearts injected with miR-19a/19b or control mimic 4 days after MI. cTNT marks cardiomyocytes (green) and DAPI marks nuclei (blue). Scale bars = 48 μm. **b**, **c** Quantification of TUNEL-positive cardiomyocytes (**b**) and non-cardiomyocytes (**c**). $n = 4$ hearts, 6–14 fields per heart. **d** qRT-PCR detection of apoptosis gene expression. $n = 3$–7 hearts. **e** Western blot analysis of protein levels of PTEN, BIM, cleaved Caspase 3, and BAX in adult hearts. **f** Quantification of Western blot band density. $n = 4$ hearts. **g** Hierarchical clustering of 544 differentially expressed genes between miR-19a/19b and control mimic injected hearts 4 days after MI. Red and blue colors indicate up-regulated or down-regulated genes. **h** Gene ontology (GO) analysis of 393 down-regulated genes. (**i**) qRT-PCR analysis of expression of down-regulated genes related to immune response, cell death, and fibrosis. Up-regulated genes were also examined. NPPA serves as a control for cardiomyopathy. $n = 3$ hearts. **j–l** Immunohistochemistry analysis of M1 markers, (**j**) CD80, (**k**) iNOS, and M2 marker (**l**) Arg-1 in adult hearts injected with miR-19a/19b or control mimics at 48 and 72 h post MI. $n = 4$–8 hearts. Scale bars = 50 μm. **m** qRT-PCR analysis of expression of M1 and M2 marker genes and potential target genes in adult hearts. $n = 3$–6 hearts. **n** Western blot analysis of protein levels of SOCS and STAT3 pathways in adult hearts. **o** Quantification of Western blot band density. $n = 3$–4 hearts. Statistical significance was calculated using Student's $t$-test in **b**, **c**, **d**, **f**, **i**, **m**, and **o** and data are presented as means ± s.e.m. *$p < 0.05$, **$p < 0.01$ vs. control. Source data are provided as a Source Data file

## Discussion

In this study, we established the functional role of miR-19a and miR-19b in protecting the heart in response to MI. Intra-cardiac injection of miR-19a/19b mimics in adult mice was sufficient to preserve cardiac function upon ischemic injury. Most importantly, systemic delivery of miR-19a/miR-19b mimics in post MI mice also produces clear cardiac protection. Given their role in cardiac protection in mouse hearts and the fact that these miR-NAs are highly conserved between mice and humans, miR-19a and miR-19b may be uniquely suited to become therapeutic targets for cardiac regeneration and heart failure.

One of the most exciting findings of this study is that miR-19a/19b appeared to protect the heart from MI injury in two phases by targeting distinct biological processes. Consistent with prior findings, we showed that miR-19a/19b enhanced cardiomyocyte proliferation in response to cardiac injury. Interestingly, direct injection of miR-19a/19b mimics or AAV9-miR-19a/19b into infarcted hearts reduced MI-induced cell death, preserving cardiac function and increasing survival. Mechanistically, our results indicate that miR-19a/miR-19b protects the heart in the early stage of heart attack by repressing the immune response in the heart. Given that recent evidence supports the critical involvement of inflammation and immune responses in cardiac protection in response to injury and remodeling, it is tempting to speculate that modulating the immune pathway brings great therapeutic promise for treating MI and heart failure. Therefore, miR-19a/19b-mediated early cardiac protection could open a window to the development of effective therapy for heart attack and bring great benefits to heart failure patients.

Numerous efforts have been applied to prevent and/or reduce cardiac injury associated with the loss of cardiomyocytes. In addition to cell therapy using stem cells or progenitor cells, understanding the molecular pathways that control cardiomyocyte proliferation yields promise for the success of gene therapy. Several miRNAs have been reported to potently enhance cell proliferation, and they have also been demonstrated to be able to promote cardiomyocyte proliferation in embryonic, postnatal, and even adult hearts. However, previous work, using transgenic overexpression approaches in mouse models, also found that long-term, continuous overexpression of microRNAs (*miR-17-92* cluster and *miR-302-367* cluster) in adult hearts resulted in a significant increase in cardiomyocyte proliferation and dedifferentiation, leading to deleterious cardiac function[10,18]. Work reported here further defines miR-19a/miR-19b as effective miRNAs that mediated the function of the *miR-17-92* cluster, suggesting that these miRNAs could become therapeutic targets for heart failure. Whereas our data establishes that miR-19a/19b are likely the key members of the *miR-17-92* cluster that mediate its function in cardiac protection, we cannot formally rule out the potential involvement of other members of this cluster.

What are the targets of miR-19a/19b in the heart? It has been generally believed that a miRNA achieves its function by targeting

many downstream mRNA targets. Interestingly, one of our recent works demonstrated that the function of one miRNA could be primarily mediated by a single target; we showed, using both gain and loss of function experiments and functional rescue assays, that miR-208a controls cardiac remodeling by repressing the expression and function of *Sox6* in the heart[27]. Here, we verified that *Pten*, one miR-19 target[18] that is involved in cell proliferation and apoptosis, is repressed by miR-19a/19b in the heart. It is likely that PTEN directly participates in miR-19a/19b-mediated regulation of cardiomyocyte proliferation in response to MI. We observed dysregulation of genes related to the immune response pathway in the early stage of cardiac response to injury. It will be important to determine whether these genes are direct targets of miR-19a/miR-19b. Most miRNAs regulate their target genes by repressing their expression levels. While we observed that many genes related to immune response were repressed by miR-19a/19b, intriguingly, we also observed up-regulated immune response genes upon miR-19a/19b overexpression, such as *Arg-1* and *CD 163*. At present it is unclear how they are up-regulated by miR-19a/19b. Better defining miR-19a/19b targets in this setting will increase our knowledge on the molecular mechanisms of miRNA action. The Hippo/Yap pathway has also been reported to play a key role in the regulation of cell proliferation and cardiac regeneration[29–33], raising the question of whether miR-19a/19b controls cardiomyocyte proliferation and heart regeneration by regulating the expression and activity of the Hippo/Yap pathway. While our study did not yield evidence to support such link, it will be interesting to investigate the potential molecular, genetic and functional interaction between miR-19 and the Hippo/Yap pathway in the heart. In particular, whether miR-19 could regulate the expression and function of the Hippo/Yap pathway post-transcriptionally.

Whereas our current study provides convincing evidence to support the positive roles of miRNAs in cardiac function, it should be recognized that certain limitations and barriers must be overcome before miRNAs can become effective therapeutic tools to treat heart disease. Among these concerns is the specificity of miRNA action. Another challenge of utilizing miRNAs is how to effectively and specifically deliver them into target sites. Nevertheless, miRNAs hold tremendous promise to become powerful tools to battle cardiovascular disease.

## Methods

**Study approvals.** The use of animals in this study conformed to the Public Health Service Guide for Care and Use of Laboratory Animals and was approved by the Institutional Animal Care and Use Committee (IACUC) of Boston Children's Hospital and by the IACUC of Zhejiang University. The human study was performed according to the principles of the Declaration of Helsinki. The Institutional Ethics Committee of the National Institute of Cardiovascular Diseases, Bratislava, Slovakia, approved the study protocol. Patients provided written informed consent.

**Human samples.** Left ventricle (LV) tissues were taken from patients with terminal-stage heart failure indicated for heart transplantation[34,35]. In brief, the

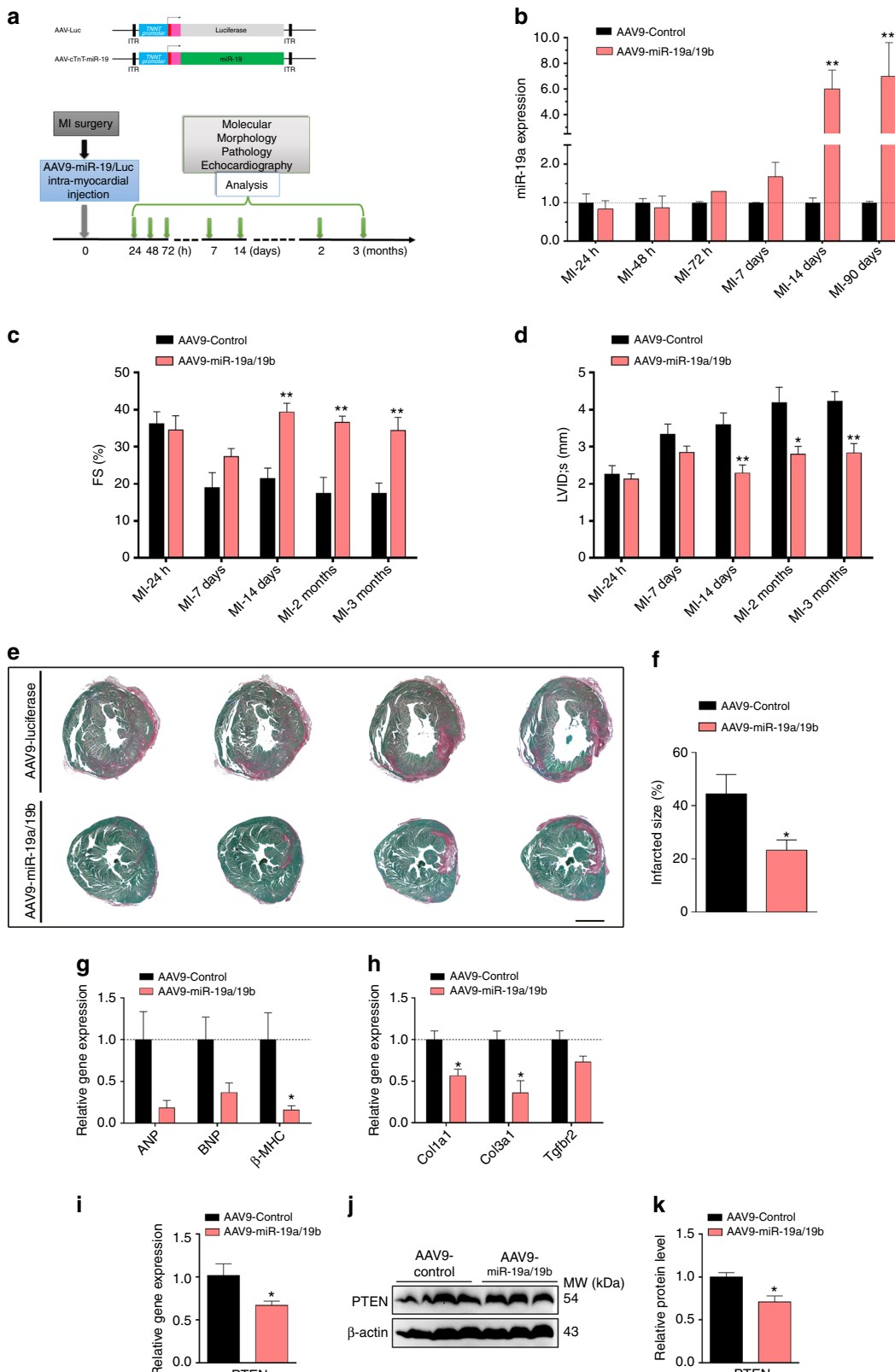

patient's heart was removed at the time of transplantation, and LV tissue was subsequently dissected and snap frozen. We used LV samples from healthy hearts that were not implanted to serve as controls.

**Myocardial infarction**. MI was performed on 8-week-old male c57BL/6 mice (Jackson Laboratory) by ligation of the LAD coronary artery. For surgery, mice are anesthetized with isoflurane (3% isoflurane for induction, 2% isoflurane for

maintenance). The chest is shaved and cleaned with alcohol. A suture is placed around the front upper incisors and pulled taut so that the neck is slightly extended. The tongue is retracted and held with forceps, and a 20-G catheter is inserted into the trachea. The catheter is then attached to the mouse ventilator via a Y-shaped connector. Ventilation is performed with a tidal volume of 225 μl for a 25 g mouse and a respiratory rate of 130 breaths per minute. 100% oxygen is provided to the inflow of the ventilator. The chest is opened through a left para-sternal incision, and the heart exposed at the left 3rd–4th intercostal space. Chest

**Fig. 6** AAV9-mediated overexpression of miR-19a/19b protects the heart from myocardial infarction. **a** Schematic of the AAV9-miR-19a/19b and AAV9-Control constructs and the experimental procedure of the AAV9-miR-19a/19b therapeutic trial in adult mice after myocardial infarction injury. ITR inverted terminal repeat. **b** qRT-PCR detection of the level of miR-19a in heart after intra-cardiac injection of AAV9-miR-19a/19b 3 months post-MI. $n = 2$–3 hearts. **c**, **d** Echocardiography analyses of cardiac function after intra-cardiac injection of AAV9-miR-19a/19b versus control 3 months post-MI. AAV9-miR-19a/19b injection (**c**) significantly increased left ventricular fractional shortening (FS%) and (**d**) decreased LV internal dimension at end-systole LVID;s. $n = 4$–7 hearts. **e** Representative images of series of transverse sections after intra-cardiac injection of AAV9-miR-19a/19b versus control at 3 months post-MI. Sirius red/fast green collagen staining marks myocardium (green) and scar (red). Scale bar = 2 mm. **f** Quantification of the scar size of the heart sections. $n = 4$ hearts. **g**–**i** qRT-PCR detection of expression of (**g**) cardiomyopathy marker genes, (**h**) fibrosis marker genes and (**i**) miR-19 target gene PTEN in hearts after intra-cardiac AAV9-miR-19a/19b injection 3 months post-MI. $n = 4$–6 hearts. **j** Western blot analysis of protein level of PTEN in hearts after intra-cardiac AAV9-miR-19a/19b injection 7 days post-MI. **k** Quantification of Western blot band density. $n = 3$ hearts. Statistical significance was calculated using Student's $t$-test in **b**, **c**, **d**, **f**, **g**, **h**, **i**, and **k** and data are presented as means ± s.e.m. *$p < 0.05$, **$p < 0.01$ vs. control. Source data are provided as a Source Data file

---

retractor is applied to facilitate the view. The pericardium is opened, and ligations made on the LAD coronary artery using 8–0 silk sutures (Ethicon). The lungs are slightly overinflated to assist in removal of air in the pleural cavity. Dissected intercostal space and chest skin were closed using a 6–0 silk suture (Ethicon).

**Intra-cardiac injection of miRNA mimics**. Eight-week-old male c57BL/6 mice were randomly subjected to intra-cardiac injection of microRNA mimic miR-19a/miR-19b or control mimics (10 μg per mouse heart), respectively, after MI. All microRNA mimics were purchased from Dharmacon and formulated with Max-Suppressor in vivo RNALancerII, a lipid-based delivery reagent (BIOO Scientific, Inc.), according to the manufacturer's instructions and a previous report[10].

For evaluation of individual members of the miR-19 family, miR-19a (10 μg per heart), miR-19b (10 μg per heart), or control mimics (10 μg/per heart) were injected post MI. For the experiments using a combination of miR-19a and miR-19b, miR-19a (5 μg)/19b (5 μg) (10 μg per heart), or control mimics (10 μg per heart) in a total volume of 50 μl were injected using an insulin syringe with needle (31 G), immediately after the ligation of LAD coronary artery. The miRNA mimics were evenly injected into three sites around the infarcted area (anterior wall, lateral wall, and apex area). Average volume of mimic solutions for each injection site is about 15 μl. When injecting, the needle is within the ventricle muscular wall but not ventricular cavity. Immediately after injection of mimics, anesthesia (isoflurane) was stopped to increase survival.

**AAV9-miR-19a/19b cloning and virus packaging**. We generated AAV9-cTNT::miR-19a/miR-19b-1(AAV9-miR-19a/19b), in which the cardiac-specific TNNT2 promoter is used to drive the expression of miR-19a/19b in the heart. An AAV9-cTNT::Luciferase (AAV9-Control) was used as a control. The cDNA fragments encoding Luciferase and mouse miR-19a/19b precursor sequences (miR-19a/19b-1 vector was gifted from Dr. Andrea Ventura[36]) were separately cloned into AAV Inverted Terminal Repeat (ITR)-containing AAV9 plasmid (Penn Vector Core) harboring the chicken cardiac TNT promoter, to generate pEn.cTnT::Luciferase and pEn.cTnT::miR-19a/19b, respectively. The AAV9 were packaged in HEK293T cells (obtained from the American Type Culture Collection and tested to be mycoplasma free) with AAV9:Rep-Cap and pAd deltaF6 (Penn Vector Core) as described previously[37]. Three days later, cells were collected and lysed. AAV was purified and concentrated by gradient centrifugation. Titration of AAV viral particles was determined by real-time PCR quantification of the number of viral genomes, as described previously[27,34]. The viral preparations had titers between $1 \times 10^{13}$ and $4 \times 10^{13}$ viral genome particles per ml.

**Intra-cardiac injection of AAV9-miRNA**. Eight week-old male c57BL/6 mice were randomly subjected to intra-cardiac injection of AAV9-miR-19a/19b, or AAV9-Control ($2 \times 10^{11}$ viral genome particles per mouse heart), respectively, after MI. Immediately after the ligation of LAD coronary artery, AAV9-miR-19a/19b or AAV9-Control in a total volume of 50 μl were injected into ventricle muscular wall but not ventricular cavity using insulin syringe with needle (31 G). The AAV9-miRNAs were evenly injected into three sites around the infarcted area (anterior wall, lateral wall, and apex area). Immediately after injection of AAV, anesthesia (isoflurane) was stopped to increase survival. Then the chest was closed and the animal was changed to a prone position until recovery of spontaneous breathing.

**Systemic delivery of miRNA mimics through tail-vein injection**. Eight-week-old male c57BL/6 mice were randomly subjected to tail-vein injection of miR-19a/19b or control mimics after MI. All miRNA mimics were purchased from Dharmacon and formulated with neutral lipid emulsion (NLE,MaxSuppressor in vivo RNA-LancerII, BIOO Scientific, Inc.) according to the manufacturer's instructions. Mice were injected a single dose per day of 10 μg NLE-formulated miRNA mimics through intravenous tail-vein. To evaluate the therapeutic potential of miR-19a/19b in treating infarcted hearts, miR-19a/19b or control mimic (10 μg in 100 μl mixture per mouse systemically) was administered daily for 3 days, beginning 6 h after MI.

To further determine the therapeutic effect of miR-19a/19b on heart after MI, another lipid-mediated (RNAiMax) miRNA delivery through tail-vein injection was performed. A dose of 100 μl mixture of miRNA and RNAiMax (ratio 1:1 in volume) was prepared. RNAiMax-mediated mixture was injected through intravenous tail-vein daily for 3 days, beginning 6 h after MI.

**Measurement of cardiac function by echocardiography**. Echocardiographic measurements were performed on mice using a Visual Sonics Vevo® 2100 Imaging System (Visual Sonics, Toronto, Canada) with a 40 MHz MicroScan transducer (model MS-550D)[18,27,34]. Mice were anesthetized with isoflurane (2.5% isoflurane for induction and 0.5% for maintenance). Heart rate and left ventricular (LV) dimensions, including diastolic and systolic wall thicknesses, LV end-diastolic and end-systolic chamber dimensions, were measured from 2-D short-axis under M-mode tracings at the level of the papillary muscle. LV mass and functional parameters such as percentage of fractional shortening (FS%) and left ventricular volume were calculated using the above primary measurements and accompanying software.

**EdU incorporation and immunofluorescence staining**. 5-ethynyl-2′-deoxyuridine (EdU)-labeling assay was used to analyze cardiomyocyte proliferation as described previously[18]. MI mice with injected miRNA mimics were intraperitoneally injected with EdU (Life Technologies) at 5 μg/g of body weight, every 2 days for five times total. At the end point of the experiment, mouse hearts were harvested, rinsed with PBS and arrested in diastole buffer (4.7 nM KCl and 0.1% 2,3-butanedione monoxime (BDM) in PBS). Hearts were then fixed in 4% paraformaldehyde (pH 8.0) overnight and embedded in paraffin. Then paraffin samples were cut into 5 μm tissue sections, de-waxed in xylene for 5 min (twice) and rehydrated with alcohols in decreasing concentration (100%, 90%, 70%, 50%) at room temperature. Sections were boiled for 10 min in antigen retrieval buffer Retrievagen A (pH 6.0) (BD Pharmingen). Sections were rinsed three times in water, permeabilized for 15 min in 0.5% Triton X100 PBS, and blocked for 30 min in 3% BSA in PBS. Then EdU Immunofluorescence staining was performed on the sections by using the Click-iT® EdU Imaging Kits (Alexa Fluor® 488) to assess EdU incorporation according to the manufacturer's instructions. For further co-staining, the sections were removed from the reaction cocktail and washed with 1 ml of 3% BSA in PBS and blocked in 5% goat serum. Mouse anti-cardiac troponin T (cTNT, 1:500, a generous gift from Dr. Jim Lin of University of Iowa) antibody was used to mark the cardiomyocytes. Nuclei were visualized with 4′,6′-diamidino-phenylindole (DAPI, 1:5000, Invitrogen). Goat anti-mouse secondary antibody (Alexa-Fluor 594, 1:400, Invitrogen) was used for visualization under microscopy. Cardiomyocyte cross-sectional area was labelled with wheat germ agglutinin (Alexa Fluor® 647 conjugate WGA; 1:100, Invitrogen), which was added together with the secondary antibody. Quantitative data were obtained by measuring co-localization of DAPI with EdU in the cardiomyocyte area. Imaging was performed on a Nikon TE2000 epifluorescent microscope with deconvolution (Volocity; Perkin-Elmer). To identify mitosis and cytokinesis, we used rabbit anti-pH3 (1:400, Millipore, cat #06-570) and rabbit anti-Aurora B (1:100, Abcam cat #ab2254) antibodies, respectively. Mouse anti-α-actinin (1:250, Abcam, cat #ab9465) was used to mark cardiomyocytes. Nuclei were visualized with DAPI (1:5000). Goat anti-rabbit and goat anti-mouse AlexaFluor 488 or 594 as secondary antibody (1:400, Invitrogen) were used for microscopic visualization. Quantitative data were obtained by measuring co-localization of DAPI (nuclear staining) with pH3 in the cardiomyocyte area.

**Histology and TUNEL assays**. Mouse hearts were dissected out, rinsed, and arrested in diastole buffer (4.7 nM KCl and 0.1% BDM in PBS). Hearts were then fixed in 4% paraformaldehyde (pH 7.4) overnight. After dehydration through a series of ethanol baths, samples were embedded in paraffin wax according to standard laboratory procedures. Sections of 5 μm in thickness were stained with

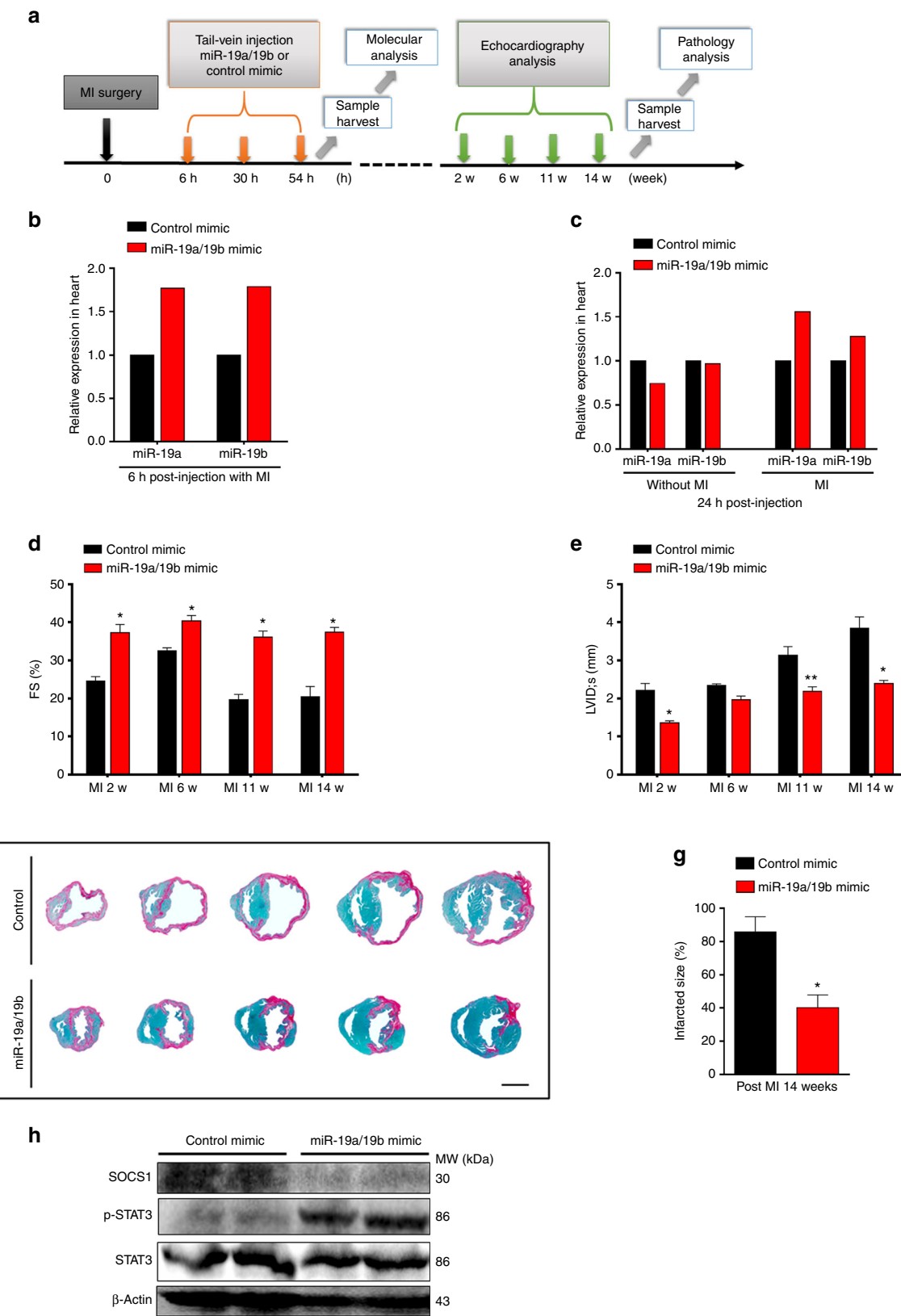

haematoxylin and eosin (H&E). The stained sections were used for routine histological examination by light microscopy.

To determine infarct size, hearts were fixed in 4% paraformaldehyde (pH 8.0), dehydrated and embedded in paraffin. Then the embedded paraffin blocks were cut through from apex to base. The first 10 sections (10 μm thickness each) of every 100 sections were used. Sections were further fixed with prewarmed Bouins' solution at 55 °C for 1 h and stained with Fast Green and Sirius Red[18,27,34]. Infarct

size was calculated according to the formula: [length of coronal infarct perimeter (epicardial + endocardial)/total LV coronal perimeter (epicardial + endocardial)] × 100 [18,38].

Terminal deoxynucleotidyl transferase-mediated nick-end labeling (TUNEL) assays were performed on paraffin sections to detect apoptotic cardiomyocytes, as described previously[18]. ApopTag® Plus In Situ Apoptosis Fluorescein Detection Kit was used (Cat #S7111 Millipore) according to the manufacturer's procedure.

**Fig. 7** Therapeutic potential of miR-19a/19b in treating infarcted hearts. **a** Schematic of tail-vein injection of RNALancerII-delivered miRNA mimics post myocardial infarction (MI). **b** qRT-PCR detection of miR-19a and miR-19b expression in the heart 6 h after injection post MI injury. $n = 1$ heart each group. **c** qRT-PCR detection of miR-19a and miR-19b expression in the heart at 24 h after injection with or without MI injury. $n = 2$ hearts each group. **d**, **e** Echocardiography analyses of cardiac function after tail-vein injection of miR-19a/19b and control mimic at different time points post MI. **d** FS%, left ventricular fractional shortening. **e** LVIDs LV internal dimension at end-systole. $n = 3$–10 mice. **f** Representative images of series of transverse heart sections after tail-vein injection of miR-19a/19b or control mimics post MI. Sirius red/Fast green collagen staining marks myocardium (green) and scar (red). **g** Quantification of the size of scar. $n = 3$ hearts. **h** Western blot analysis of protein levels of SOCS and STAT3 in the heart 24 h after tail-vein injection of miR-19a/19b or control mimics post MI. Statistical significance was calculated using Student's $t$-test in **d**, **f**, and **g** and data are presented as means ± s.e.m. *$p < 0.05$, **$p < 0.01$ vs. control. Source data are provided as a Source Data file

Positive control slides containing unstained rat mammary glands were TUNEL assayed as well. The cardiomyocytes were counter-stained with cTNT and DAPI.

To identify M1 and M2 phages, immunohistochemistry analyses were performed in the heart sections. After incubating paraffin heart sections in three washes of xylene, two washes of 100% ethanol, two washes of 95% ethanol, and two washes of ddH$_2$O for deparaffinization and rehydration, they were processed for antigen unmasking according to the manufacturer's protocol. Primary and secondary antibodies were as follows: CD80 (1:50, M1007-10, Hangzhou Huaan Biotechnology Co., Ltd.), iNOS (1:200, RT1332, Hangzhou Huaan Biotechnology Co., Ltd.), Arginase 1 (1:200, RT1051, Hangzhou Huaan Biotechnology Co., Ltd.). UltraVision HP IHC detection kit was obtained from Shanghai Universal Biotech Co., Ltd. Pictures were taken with a Leica DM4000 microscope.

**Western blot analysis**. Protein lysate samples were prepared from heart tissues in tissue extraction reagent (Invitrogen, FNN0071) supplemented with proteinase inhibitors. Lysate samples (30 μg total protein for each) were separated by 10% SDS–PAGE and electrophoretically transferred to PVDF membranes. SOCS1 protein was probed with goat antibody to SOCS1 (Abcam, ab9870; 1:1000 dilution). SOCS3 protein was probed with rabbit antibody to SOCS3 (Abcam, ab16030; 1:1000 dilution). STAT3 protein was probed with mouse antibody to STAT3 (CST, #9139; 1:1000 dilution). Phospho-STAT3 protein was probed with rabbit antibody to phosho-STAT3 (CST, #9145; 1:1000 dilution). PTEN protein was probed with rabbit antibody to PTEN (CST, #9188; 1:1000 dilution). Bim protein was probed with rabbit antibody to Bim (CST, #2933; 1:1000 dilution). Cleaved-caspase3 protein was probed with rabbit antibody to cleaved-caspase3 (CST, #9661; 1:1000 dilution). Bcl2 protein was probed with rabbit antibody to Bcl2 (CST, #3498; 1:1000 dilution). Bax protein was probed with rabbit antibody to Bax (CST, #2772; 1:1000 dilution). Protein bands were visualized with the Bio-Rad ChemiDoc imaging system. All the antibody information is listed in Supplementary Table 6.

**RNA-seq and genome-wide transcriptome analysis**. RNA from the hearts of mice 4 days after MI and miRNA mimic injection was prepared for RNA-seq (three biological replicates for each group). RNA-seq experiments were performed by Novogene (Beijing, China). Briefly, total RNA was isolated from fresh ventricular tissue using TRIzol (Invitrogen). mRNA was then purified from total RNA using poly-T oligo-attached magnetic beads. Sequencing libraries were generated using NEBNext® UltraTM RNA Library Prep Kit for Illumina® (NEB, USA) following manufacturer's recommendations, and index codes were added to attribute sequences to each sample. The clustering of the index-coded samples was performed on a cBot Cluster Generation System using TruSeq PE Cluster Kit v3-cBot-HS (Illumia) according to the manufacturer's instructions. After cluster generation, the library preparations were sequenced on an Illumina Hiseq platform and 150 bp paired-end reads were generated. For the data analysis, raw data (raw reads) in fastq format were first processed through in-house Perl scripts. Clean data (clean reads) were obtained by removing reads containing adapters, reads containing ploy-N and low-quality reads from raw data. Reference genome and gene model annotation files were downloaded from genome website directly. Index of the reference genome was built using STAR and paired-end clean reads were aligned to the reference genome using STAR (v2.5.1b). STAR uses the method of Maximal Mappable Prefix (MMP). HTSeq v0.6.0 was used to count the read numbers mapped to each gene. Analysis of differential expression was performed using the edgeR R package (3.12.1). The $P$ values were adjusted using the Benjamini and Hochberg method. GO and KEGG pathway analyses were implemented using the clusterProfiler R package. The hierarchical clustering heat map was generated with the ggplot library.

**Isolation of cardiomyocytes from adult mouse hearts**. Isolation of cardiomyocytes was performed as described previously[39–41]. Briefly, fresh hearts were harvested and immediately fixed in 4% paraformaldehyde for 2 h at room temperature. Samples were then washed in PBS for 5 min and digested with collagenase B (1.8 mg/ml, Roche) and collagenase D (2.4 mg/ml, Roche) on a rotator at 37 °C overnight. Cell suspensions were then collected at 300 g for 2 min and the pelleted cells were resuspended in PBS. The isolated cardiomyocytes were stained with Connexin 43 antibody (1:1000, Abcam, cat #ab11370), α-actinin (1:250, Abcam, cat #ab9465),

and DAPI (1:5000) for cardiomyocyte counts and nucleation counts. For cardiomyocyte count, 10 μl aliquots samples from total volume of digested cardiomyocytes were counted using hemocytometer. The total number of cardiomyocytes counted was ~150–200 cardiomyocytes per aliquot. We averaged three different counts per sample and eight hearts per group. For nucleation, ~1 × 10$^3$ cardiomyocytes were counted per sample, using six independent samples per group. Nucleation was plotted as percentage of counted cardiomyocytes.

**Quantitative RT-PCR**. Total RNA was isolated using TRIzol Reagent (Invitrogen) from cell or tissue samples. For qRT-PCR detecting the expression of protein-coding genes, 2.0 μg RNA samples were reverse-transcribed to cDNA using random hexamers and MMLV reverse transcriptase (Invitrogen) in a 20 μl reaction system. In each analysis, 0.1 μl cDNA pool was used for quantitative PCR. For qRT-PCR detecting the expression of miRNAs, 10 ng RNA samples were reverse-transcribed to cDNA using TaqMan® MicroRNA Reverse Transcription Kit (ABI). In each analysis, 1.5 μl cDNA pool and TaqMan® MicroRNA Assays were used for quantitative PCR. All qPCR experiments were performed on the Applied Biosystems 7500 Real-Time PCR System. Primers used in this study is listed in Supplementary Table 7.

**Reporting summary**. Further information on experimental design is available in the Nature Research Reporting Summary linked to this article.

## Data availability

All relevant data related to this manuscript are available on request from the authors on reasonable request. The accession number for the RNA-sequencing data described in this study is GSE126888. Original un-cropped western blots are provided in source data. The source data underlying Figs. 1–7 and Supplementary Figs. 2–5 are provided.

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

## Acknowledgements

We thank members of the Wang and Chen laboratories for advice and support. We thank Dr. Andrea Ventura (Memorial Sloan Kettering Cancer Center) for providing mouse miR-19a/19b plasmid. This research is supported by National Key R&D Program of China (2017YFA0103700), Grants from National Natural Science Foundation of China (Nos. 81470382, 81670257 for J.C.). Work in the Wang lab is supported by American Heart Association, Muscular Dystrophy Association, and the NIH (HL085635, HL116919, HL138757). M.K. was supported by Banyu Life Science Foundation International. Z.-P.H. was supported by NIH T32HL007572 and American Heart Association Scientist Development Grant (SDG). J.D. was supported by NIH T32HL007572.

## Author contributions

J.C. and D.-Z.W. conceived of and supervised the study. F. Gao, J.C., and D.-Z.W. designed the experiments and analyzed the data, and wrote the manuscript. M.K., Q.M., and Y.W. performed myocardial infarction surgery model on mice. F. Gao, M.K. and J.C. performed echocardiographic data acquisition and analysis. F. Gao, N.L., T.L., Z.-P.H., J. D., J.L., F.Z., M.Z., Xiaoyun Hu, and J.C. performed molecular biology experiments and contributed to the histological and immunofluorescent data acquisition and analysis. F. Gao, Z.-P.H., F. Gu and J.C. performed RNA-seq experiments and data analysis. N.L. and J.D. contributed to adeno-associated virus preparation. J.K. contributed to human sample acquisition. Xinyang Hu, W.T.P. and J.W. refined the data analysis and reviewed the manuscript.

## Additional information

**Competing interests:** The authors declare no competing interests.

