## [Peer Review File · Nature Communications]

Reviewers' comments:

Reviewer #1 (Remarks to the Author):

Myocardial infarction results in significant loss of cardiomyocytes and the inability to regenerate the affected tissues. This manuscript describes a focused and nicely conducted study regarding the therapeutic role of miR-19a/19b in both cardiac regeneration and protection immediately after myocardial infarction is induced. The results are quite convincing that miR19a manipulation induces improvement in ventricular function. Previously, the same group has shown that miR-17-92 cluster is required for and sufficient to induce cardiomyocytes proliferation in postnatal and adult hearts (Chen et al., *Cir Res* 2013) using the same animal models. This is a follow-up study focusing on a specific miRNA within the miR-17-92 cluster.

1. Ouchida et al (*Plos one*, 2012) reported that miR-20a has a more prominent proliferative effect than miR-19a. The author should explain how miR-19a/b were chosen as major contributors on cell proliferation since the same group showed that miR-17-92a cluster had a proliferative effect in the previous publication.
2. Bim is a direct target of miR-19a in previous literature (Gao et al., *Toxi Letter*, 2016). In addition, the authors as well as other groups have already shown that miR-19a has anti-apoptotic activities. It is highly suggested that major signaling molecules relating apoptosis are shown. Alternative methods to show protein expression including western blot will strengthen the article.
3. Clinically, this method of applying miR-19a/19b at the same time of a myocardial infarction is unlikely. Therefore, instead of being therapeutic, this method seems to have more of a preventative aspect. If the authors want to state THERAPEUTIC applications, functional analysis would be required before injection for gene of interest.

Reviewer #2 (Remarks to the Author):

The current manuscript builds upon a previous publication by Chen et al (*Circ Res*, 2013), which showed that the miR-17-92 cluster was necessary and sufficient for induction of cardiomyocyte proliferation. The authors have now extended their findings to demonstrate that two members of this microRNA cluster, miR-19a/b, are sufficient to induce adult cardiomyocyte proliferation post-MI. The cardioprotective effects of intracardiac delivery of miR-19a/b appear to be multi-faceted and include actions on the immune response, fibrotic response, myocyte survival and myocyte proliferation. The authors demonstrate that a single intramyocardial injection of miR-19a/b is

sufficient to improve cardiac function and reduce fibrosis long-term. The findings are novel and could have important therapeutic implications for the treatment of ischemic heart disease and heart failure. However, some aspects of the data set are difficult to interpret and require further clarification.

Major Criticisms:

1. miR-19a/19b mimic and AAV-miR-19a/19b: It is unclear why both members of the miR-19 family were concurrently delivered to the adult heart. Why weren't mimics/AAVs expressing individual miR-19 family members delivered (i.e. miR-19a or miR-19b)? Mimic sequences should also be provided along with additional methodological details regarding the protocol for complexing multiple microRNA mimics (i.e. amount of each mimic (10ug total or 10ug of each mimic?), ratio of lipofectamine to RNA mimic, etc).
2. Post-MI mortality rate: A mortality rate of 50% for the mice injected with control microRNA mimic following MI seems unusually high for this procedure. Please provide literature support for similar mortality rates in the C57BL/6 strain or otherwise account for why the mortality rate is so high (e.g. position of LAD ligature, intramyocardial injection (needle trauma) or is this due to toxicity of the control microRNA mimic?).
3. Kinetics of miR over-expression: In order to understand the physiological effects of the microRNA, the kinetics of miR expression need to be more fully characterised in the mimic (Figure 2B) and AAV (Figure 6B) injection groups. Additional early time points (day 1, day 7 and day 14) should be included. This is particularly important for interpretation of the AAV results because it is currently unclear whether the AAV-mediated (CM-specific) delivery is also having an acute cardioprotective effects. The authors should refer to a recent study by Lesizza et al (PMID: 28077443) for supporting data regarding the time-course of microRNA over-expression following intracardiac injection of mimics.
4. Early versus late effects: In my opinion, all of the results could be explained by an early (acute) cardioprotective effect of the microRNA post-MI. If miR-19 reduces infarct size (e.g. reduces cell death immediately following MI), myocardial mass would be preserved and there would subsequently be less fibrosis and improved heart function long-term. Therefore, there is no evidence that the microRNA exerts its effects in two phases. The authors should directly assess infarct size (by tetrazolium chloride staining) at 24 hours post-MI. It would also be helpful if heart function (echo) was performed at baseline and again at early time points post-MI (e.g. day 1, day 3, day 7) when acute protective effects of the miR mimic might be observed.
5. AAV-miR-19 (mechanistic comparison to mimic): The AAV results suggest that a similar cardioprotective effect is obtained when miR-19 is delivered specifically to cardiomyocytes. This experiment allows the authors to tease apart mechanisms of action compared with the non-cell-type-specific mimic, which had pleiotropic effects on the inflammatory response, fibrosis and CM survival/proliferation. What is the time course of AAV-mediated miR-19 expression post-MI compared with mimic (see above)? Does the AAV-miR-19 acutely protect the heart (i.e. reduced

infarct size (TTC) at 24h)? Does the AAV-miR-19 also reduce inflammatory and fibrotic markers? If so, these effects likely occur secondary to cardiomyocyte-specific actions of miR-19.

Other:

1. Figure 1D&1E: There should be a sham control for each time point.
2. Figure 1F: The significance of the DOX data is unclear. Given that this result appears to be from n=1 experiment, the data should be removed. In my opinion, it does not add anything to the manuscript.
3. Given that miRNA target genes have not been fully characterised in this study and the effects of miR-19 appear to be complex (potentially involving multiple cell types), the discussion needs to be toned down. Specifically, the authors state that "Given that most miRNA direct regulated target genes are repressed by their miRNAs and we observed upregulated immune response genes upon miR-19a/19b overexpression, these genes are unlikely direct miR-19a/19b targets." This statement is factually incorrect (immune response genes were the most significantly DOWN-REGULATED set of genes according to Figure 5F), so the results could, in part, be accounted for by a direct action of miR-19 in immune cells. Further mechanistic studies are required to resolve this complex issue, which is probably beyond the scope of the current study.

Reviewer #3 (Remarks to the Author):

The authors' major finding is that over expression of the miRs 19a/b "protects the rat heart" from experimentally- created myocardial infarction" and stimulate cardiogenesis and improve the function of the heart as a result. Expression of these miRs also appears to diminish the expression of genes that promote inflammation and apoptosis . The experimental work appears to have been well done. However, there are some issues that need to be addressed. The main issue that has been ignored in this work is previous work done by Dr. James Martin and his team showing that the murine heart expresses a Stop Growth Pathway within 2 weeks of life, the Hippo Pathway, a kinase cascade

that interacts with YAP and Park 2 to regulate the growth and regenerative capability of the heart. When Hippo is activated around 2 weeks of life, the heart subsequently has limited ability to regenerate itself after injury , including myocardial infarction, but when Hippo is silenced at the time of the MI

or inhibited 3 weeks after MI, the murine heart will very nearly completely regenerate the area of injury, reduce the scar size, improve the function of the injured heart, and promote angiogenesis as

the result of activating Yap and Park 2{ Refs. Heallen,T et al Development 2013;140(23)4683-4690; Morikawa et al Science Signaling 2015 ;8(375):pc11; Martin JF et al Circ Res 2017;121:13-15; and a paper in press in Nature , 2017 Leach JP

et al" Hippo Pathway Deletion Reverses Systolic Heart Failure" that need to be referenced and considered in this work. Do these miRs act through inhibiting Hippo and or promoting Yap and Park 2 activity? If they do, the results described here represent another means to activate Yap and Park 2 rather than a novel way to stimulate cardiogenesis. There are other issues too, including is angiogenesis stimulated by these miRs in the rat model with MI ? The authors indicate that these miRs increase in the infarcted rat heart without being protective so is it necessary to over express these miRs to see the protection the authors have shown in this work? ; and their Ref 10 describes some interaction between other miRs and Hippo -Yap, but this is never considered again in their work with the miRs 19a/b .

Response to reviewers' comments:

Reviewer #1 (Remarks to the Author):

Myocardial infarction results in significant loss of cardiomyocytes and the inability to regenerate the affected tissues. This manuscript describes a focused and nicely conducted study regarding the therapeutic role of miR-19a/19b in both cardiac regeneration and protection immediately after myocardial infarction is induced. The results are quite convincing that miR19a manipulation induces improvement in ventricular function. Previously, the same group has shown that miR-17-92 cluster is required for and sufficient to induce cardiomyocytes proliferation in postnatal and adult hearts (Chen et al., Cir Res 2013) using the same animal models. This is a follow-up study focusing on a specific miRNA within the miR-17-92 cluster.

1. Ouchida et al (Plos one, 2012) reported that miR-20a has a more prominent proliferative effect than miR-19a. The author should explain how miR-19a/b were chosen as major contributors on cell proliferation since the same group showed that miR-17-92a cluster had a proliferative effect in the previous publication.

Response: We are aware of the Ouchida report and recognize that their study is in a different context (in breast cancer cells). We have chosen to focus on miR-19a/b because our prior study indicated that miR-19 potently regulates cardiomyocyte proliferation (Chen et al., Circ. Res. 112: 1557, Figure 5). We have now further clarified this issue in the revision (Page 3).

2. Bim is a direct target of miR-19a in previous literature (Gao et al., Toxi Letter, 2016). In addition, the authors as well as other groups have already shown that miR-19a has anti-apoptotic activities. It is highly suggested that major signaling molecules relating apoptosis are shown. Alternative methods to show protein expression including western blot will strengthen the article.

Response: We have examined the expression level of Bim protein and found it is repressed by miR-19a/19b. In addition, we examined the expression of several other molecules related to the apoptosis pathway, including caspase 3, and found they are regulated in miR-19 mimic injected hearts. These new data are now included in the new Figure 5 and Supplementary Figure 3 of the revision.

3. Clinically, this method of applying miR-19a/19b at the same time of a myocardial infarction is unlikely. Therefore, instead of being therapeutic, this method seems to have more of a preventative aspect. If the authors want to state THERAPEUTIC applications, functional analysis would be required before injection for gene of interest.

Response: We have now directly tested the therapeutic applications of miR-

19a/19b according to the suggestions. As shown in the new Figure 7, supplementary Figure 4, and supplementary table 5, we found that systemic delivery of miR-19a/19b mimics via tail-vein injection in post myocardial infarction (MI) mice significantly reduced MI-mediated cardiac injury.

Reviewer #2 (Remarks to the Author):

The current manuscript builds upon a previous publication by Chen et al (Circ Res, 2013), which showed that the miR-17-92 cluster was necessary and sufficient for induction of cardiomyocyte proliferation. The authors have now extended their findings to demonstrate that two members of this microRNA cluster, miR-19a/b, are sufficient to induce adult cardiomyocyte proliferation post-MI. The cardioprotective effects of intracardiac delivery of miR-19a/b appear to be multi-faceted and include actions on the immune response, fibrotic response, myocyte survival and myocyte proliferation. The authors demonstrate that a single intramyocardial injection of miR-19a/b is sufficient to improve cardiac function and reduce fibrosis long-term. The findings are novel and could have important therapeutic implications for the treatment of ischemic heart disease and heart failure. However, some aspects of the data set are difficult to interpret and require further clarification.

Major Criticisms:

1. miR-19a/19b mimic and AAV-miR-19a/19b: It is unclear why both members of the miR-19 family were concurrently delivered to the adult heart. Why weren't mimics/AAVs expressing individual miR-19 family members delivered (i.e. miR-19a or miR-19b)? Mir mimic sequences should also be provided along with additional methodological details regarding the protocol for complexing multiple microRNA mimics (i.e. amount of each mimic (10ug total or 10ug of each mimic?), ratio of lipofectamine to RNA mimic, etc).

Response: We used miR-19a/19b in our experiments because our pilot experiments showed that both miR-19a and miR-19b regulates cardiac function in a comparable manner. We have now provided new data to show the function of individual miR-19 family members (miR-19a or miR-19b) in the revision (Fig. 2C, 2D; Supplementary Table 1). We have revised the materials and methods section and included detailed information about the miR-19a and miR-19b mimics and the dosages used.

2. Post-MI mortality rate: A mortality rate of 50% for the mice injected with control microRNA mimic following MI seems unusually high for this procedure. Please provide literature support for similar mortality rates in the C57BL/6 strain or otherwise account for why the mortality rate is so high (e.g. position of LAD

ligature, intramyocardial injection (needle trauma) or is this due to toxicity of the control microRNA mimic?).

Response: In our initial MI experiments, we purposely induced large myocardial infarction by ligating the LAD at a pretty high position; in that way, we aimed to test the potential protective role of miR-19. That is probably why we got a high mortality rate. Similar high mortality rate was reported previously, which we cited in the revision (Orlic et al., PNAS 2001; Lesizza et al., Circ. Res 2017). We have gone back and performed new experiments where we reduced the severity of infarction. Accordingly, we observed lower mortality rate (~20-30%). Consistent with our previous results, we found that injection of miR-19a/19b mimics protected cardiac function and reduced the mortality rate. These new data are now included in the Suppl. Figure 2A.

3. Kinetics of miR over-expression: In order to understand the physiological effects of the microRNA, the kinetics of miR expression need to be more fully characterised in the mimic (Figure 2B) and AAV (Figure 6B) injection groups. Additional early time points (day 1, day 7 and day 14) should be included. This is particularly important for interpretation of the AAV results because it is currently unclear whether the AAV-mediated (CM-specific) delivery is also having an acute cardioprotective effects. The authors should refer to a recent study by Lesizza et al (PMID: 28077443) for supporting data regarding the time-course of microRNA over-expression following intracardiac injection of mimics.

Response: Thank you for the suggestion. We have now performed these experiments, and the new data are shown in Figure 6 and Supplementary Figure 1. We observed that miRNA mimics are taken up by cardiomyocytes ~6-12 h after intra-cardiac injection (Supplementary Figure 1). For AAV-miRNA injection, we detected increased miRNA levels at ~72 h after injection. Increased miRNA expression could still be detected in the hearts 90 days after AAV injection (Figure 6B). We have also cited Lesizza et al., Circ. Res 2017 and added additional discussion accordingly in the revision.

4. Early versus late effects: In my opinion, all of the results could be explained by an early (acute) cardioprotective effect of the microRNA post-MI. If miR-19 reduces infarct size (e.g. reduces cell death immediately following MI), myocardial mass would be preserved and there would subsequently be less fibrosis and improved heart function long-term. Therefore, there is no evidence that the microRNA exerts its effects in two phases. The authors should directly assess infarct size (by tetrazolium chloride staining) at 24 hours post-MI. It would also be helpful if heart function (echo) was performed at baseline and again at

early time points post-MI (e.g. day 1, day 3, day 7) when acute protective effects of the miR mimic might be observed.

Response: We have now performed these experiments. As shown in new Figures 2 and 6 and Supplementary tables 1 and 3, we found that 5 days after intra-cardiac injection of miRNA mimics, cardiac function is improved (Figure 2C); in contrast, it took longer time to exhibit substantial cardiac protection when AAV-miRNAs were injected; there was no difference at 24 h, but at 7 days there was improved cardiac function by AAV-19a/19b. However, they are not statistically significant (Figure 6C). These observations are consistent with the pharmacokinetics of miRNA expression in cardiomyocytes (Figure 6B; Supplemental Figure 1).

5. AAV-miR-19 (mechanistic comparison to mimic): The AAV results suggest that a similar cardioprotective effect is obtained when miR-19 is delivered specifically to cardiomyocytes. This experiment allows the authors to tease apart mechanisms of action compared with the non-cell-type-specific mimic, which had pleiotropic effects on the inflammatory response, fibrosis and CM survival/proliferation. What is the time course of AAV-mediated miR-19 expression post-MI compared with mimic (see above)? Does the AAV-miR-19 acutely protect the heart (i.e. reduced infarct size (TTC) at 24h)? Does the AAV-miR-19 also reduce inflammatory and fibrotic markers? If so, these effects likely occur secondary to cardiomyocyte-specific actions of miR-19.

Response: As discussed above and described in the revision, we did observe a delay in the protection of cardiac function when comparing AAV-cTNT-miR-19a/19b vs mimic injection. We attribute such difference to distinct miRNA delivery methods and the dynamic expression and function patterns of introduced miRNAs in the heart. At present, we don't have evidence to rule out the possibility that such difference is due to the function of miRNAs in different cell types of the heart.

Other:

1. Figure 1D&1E: There should be a sham control for each time point.

Response: We have now included sham controls in this study (Figure 1).

2. Figure 1F: The significance of the DOX data is unclear. Given that this result appears to be from n=1 experiment, the data should be removed. In my opinion, it does not add anything to the manuscript.

Response: This data is removed. Thank you.

3. Given that miRNA target genes have not been fully characterised in this study and the effects of miR-19 appear to be complex (potentially involving multiple cell types), the discussion needs to be toned down. Specifically, the authors state that “Given that most miRNA direct regulated target genes are repressed by their miRNAs and we observed upregulated immune response genes upon miR-19a/19b overexpression, these genes are unlikely direct miR-19a/19b targets.” This statement is factually incorrect (immune response genes were the most significantly DOWN-REGULATED set of genes according to Figure 5F), so the results could, in part, be accounted for by a direct action of miR-19 in immune cells. Further mechanistic studies are required to resolve this complex issue, which is probably beyond the scope of the current study.

Response: Thank you for pointing this out and we agree. We have rephrased it into “Most miRNAs regulate their target genes by repressing their expression levels. While we observed that many genes related to immune response were repressed by miR-19a/19b, intriguingly, we also observed up-regulated immune response genes upon miR-19a/19b overexpression, such as Arg-1 and CD 163. At present it is unclear how they are up-regulated by miR-19a/19b. Better defining miR-19a/19b targets in this setting will increase our knowledge on the molecular mechanisms of miRNA action.” in the revision (page 11, discussion).

Reviewer #3 (Remarks to the Author):

The authors' major finding is that over expression of the miRs 19a/b "protects the rat heart" from experimentally- created myocardial infarction" and stimulate cardiogenesis and improve the function of the heart as a result. Expression of these miRs also appears to diminish the expression of genes that promote inflammation and apoptosis. The experimental work appears to have been well done. However, there are some issues that need to be addressed.

The main issue that has been ignored in this work is previous work done by Dr. James Martin and his team showing that the murine heart expresses a Stop Growth Pathway within 2 weeks of life, the Hippo Pathway, a kinase cascade that interacts with YAP and Park 2 to regulate the growth and regenerative capability of the heart. When Hippo is activated around 2 weeks of life, the heart subsequently has limited ability to regenerate itself after injury , including myocardial infarction, but when Hippo is silenced at the time of the MI or inhibited 3 weeks after MI, the murine heart will very nearly completely regenerate the area of injury, reduce the scar size, improve the function of the injured heart, and promote angiogenesis as the result of activating Yap and Park 2{ Refs. Heallen,T et al Development 2013;140(23)4683-4690; Morikawa et al Science Signaling 2015 ;8(375):pc11; Martin JF et al Circ Res 2017;121:13-15; and a paper in

press in Nature , 2017 Leach JP et al" Hippo Pathway Deletion Reverses Systolic Heart Failure" that need to be referenced and considered in this work. Do these miRs act through inhibiting Hippo and or promoting Yap and Park 2 activity? If they do, the results described here represent another means to activate Yap and Park 2 rather than a novel way to stimulate cardiogenesis. There are other issues too, including is angiogenesis stimulated by these miRs in the rat model with MI ? The authors indicate that these miRs increase in the infarcted rat heart without being protective so is it necessary to over express these miRs to see the protection the authors have shown in this work? ; and their Ref 10 describes some interaction between other miRs and Hippo -Yap, but this is never considered again in their work with the miRs 19a/b .

Response: The Hippo/Yap pathway is clearly an essential regulator of cardiomyocyte proliferation and cardiac regeneration. We are very aware of Dr. Martin's work. In fact, we looked at the potential regulation of the Hippo/Yap pathway by miR-19a/19b. In our unbiased transcriptome analysis (Figure 5), the Hippo/Yap pathway did not appear as a main pathway regulated. As shown below, we examined the expression levels of components of the Hippo/Yap pathway and noticed that most of them were not significantly altered in miR-19a/19b mimic treated hearts. Of course, this result cannot formally rule out the possibility that the Hippo/Yap pathway is still involved in miR-19a/19b-mediated regulation of cardiomyocyte proliferation. Nevertheless, we have cited related papers and added discussion in the revision (page 11).

Expression of the Hippo/Yap pathway in miR-19a/19b mimic treated hearts

Reviewers' comments:

Reviewer #2 (Remarks to the Author):

The authors have comprehensively addressed all of the concerns raised in my initial review. This is a well-executed study and the new data substantially strengthen the overall conclusions of the study. The authors should be congratulated on this important contribution to the field.

Reviewer #3 (Remarks to the Author):

I am disappointed that the authors did not respond to Reviewer No. 3's concerns fully. I requested that they determine whether miR 19 a and miR 19b act through the Hippo Yap pathway or perhaps stimulate Yap directly, as it is already known this is an important pathway for cardiac regeneration. If this is the case, the information here would not be novel, but confirmatory of earlier work. In response to my suggestion, they have acknowledged

the earlier work and that this does indeed need to be examined in the future. I believe they need to examine it as part of the present paper as I have indicated here.

Reviewers' comments:

Reviewer #2 (Remarks to the Author):

The authors have comprehensively addressed all of the concerns raised in my initial review. This is a well-executed study and the new data substantially strengthen the overall conclusions of the study. The authors should be congratulated on this important contribution to the field.

Response: Thank you! To better strength the conclusion of miR-19a/19b enhanced cardiomyocyte proliferation and cardiac regeneration, we have now performed new experiments. In particular, we directly counted cardiomyocyte numbers and assessed the nucleation status in mouse hearts. Our data, as presented in the revised Figure 4, demonstrate that injection of miR-19a/19b mimics increased total cell numbers of cardiomyocytes three weeks after myocardial infarction (Figure 4I, J). We observed an increase of mono-nuclear cardiomyocytes and decrease of bi-nucleated cardiomyocytes (Figure 4 K, L).

Reviewer #3 (Remarks to the Author):

I am disappointed that the authors did not respond to Reviewer No. 3's concerns fully. I requested that they determine whether miR 19 a and miR 19b act through the Hippo Yap pathway or perhaps stimulate Yap directly, as it is already known this is an important pathway for cardiac regeneration. If this is the case, the information here would not be novel, but confirmatory of earlier work. In response to my suggestion, they have acknowledged the earlier work and that this does indeed need to be examined in the future. I believe they need to examine it as part of the present paper as I have indicated here.

Response: We thank this reviewer for the insightful view about the potential molecular and functional interaction between miR-19a/19b and the Hippo/Yap pathway during cardiomyocyte proliferation and cardiac regeneration. We did not observe such interaction in our investigation. These results are now presented in Supplemental Figure 3. We fully understand that this result cannot formally rule out the possibility that the Hippo/Yap pathway is still involved in miR-19a/19b-mediated regulation of cardiomyocyte proliferation, and we have now further discussed this point in the revised manuscript. We agree that this remains an important question for future investigation.

REVIEWERS' COMMENTS:

Reviewer #2 (Remarks to the Author):

The authors have addressed all of my concerns. New data has been provided showing a clear increase in cardiomyocyte number and proportion of mononucleated cardiomyocytes, which strengthens the authors' overall conclusions regarding the impact of miR-19 on cardiomyocyte proliferation.